# C–N bond formation by a polyketide synthase

Jialiang Wang [1,5], Xiaojie Wang[1,2,5], Xixi Li[1,5], LiangLiang Kong[3,5], Zeqian Du[1,5], Dandan Li [1,5], Lixia Gou[4], Hao Wu[1], Wei Cao[1], Xiaozheng Wang[1], Shuangjun Lin [1] ✉, Ting Shi [1] ✉, Zixin Deng [1] ✉, Zhijun Wang [1] ✉ & Jingdan Liang [1] ✉

Assembly-line polyketide synthases (PKSs) are molecular factories that produce diverse metabolites with wide-ranging biological activities. PKSs usually work by constructing and modifying the polyketide backbone successively. Here, we present the cryo-EM structure of CalA3, a chain release PKS module without an ACP domain, and its structures with amidation or hydrolysis products. The domain organization reveals a unique "∞"-shaped dimeric architecture with five connected domains. The catalytic region tightly contacts the structural region, resulting in two stabilized chambers with nearly perfect symmetry while the N-terminal docking domain is flexible. The structures of the ketosynthase (KS) domain illustrate how the conserved key residues that canonically catalyze C–C bond formation can be tweaked to mediate C–N bond formation, revealing the engineering adaptability of assembly-line polyketide synthases for the production of novel pharmaceutical agents.

Polyketide synthases (PKSs) are multifunctional enzymes that synthesize diverse bioactive natural products[1], many of which are widely valued as medicines[2], such as antibiotics (e.g., erythromycin and amphotericin B), cholesterol-lowering drugs (e.g., lovastatin and semi-synthetic simvastatin), antitumor agents (e.g., doxorubicin and epothilone) and immunosuppressants (e.g., rapamycin and FK-506) (Fig. 1a, b). Accordingly, there is a critical need to continue to enlarge this class of molecules by mining and, better yet, engineering more diverse compounds and hybrids toward different diseases[3].

Type 1 modular PKSs are multidomain megaenzymes that work in an assembly line fashion, and rational engineering designs have been successively introduced in steps in the synthesis process for the further diversification of complex polyketides[4–6]. As such, structural dissection of these protein machines will not only lay the grounds for understanding polyketide extension and modification but also provide indispensable templates for domain and/or module design for the production of novel pharmaceutical agents[6].

The polyether antibiotic calcimycin (A23187) in *Streptomyces chartreusis* NRRL 3882 is widely used as a biochemical tool in pharmacological and in vitro toxicological studies[7]. As a divalent cation ionophore[8], calcimycin is also able to uncouple oxidative phosphorylation of mammalian cells[9], inhibit ATPase activity[10] and induce apoptosis via activation of intracellular signaling[11]. Structurally, calcimycin is composed of an α-ketopyrrole, a spiroketal ring (i.e., a polyketide) which is synthesized by the above-mentioned type I modular PKSs, and a substituted benzoxazole moiety which involves an unusual C–N bond formation and product release[12] (Supplementary Fig. 1). Reported by X-ray crystallographic studies[13,14], the C–N bond contained benzoxazole system is crucial for calcium ion binding and thus indispensable for this ionophore-mediated biological processes. However, the mechanism by which this unique amidation reaction and chain release in calcimycin biosynthesis remains unknown.

Current structural knowledge of full PKSs[15–20] and mammalian fatty acid synthase[21–23] has provided great insights into the domain

[1]State Key Laboratory of Microbial Metabolism, Joint International Research Laboratory of Metabolic & Developmental Sciences, and School of Life Sciences and Biotechnology, Shanghai Jiao Tong University, Shanghai, China. [2]Department of Molecular Biology, Shanghai Jikaixing Biotech Inc., Shanghai 200131, China. [3]National Facility for Protein Science in Shanghai, Chinese Academy of Sciences, Shanghai 201204, China. [4]School of Life Science, North China University of Science and Technology, Tangshan, Hebei, China. [5]These authors contributed equally: Jialiang Wang, Xiaojie Wang, Xixi Li, LiangLiang Kong, Zeqian Du, Dandan Li. ✉e-mail: linsj@sjtu.edu.cn; tshi@sjtu.edu.cn; zxdeng@sjtu.edu.cn; wangzhijun@sjtu.edu.cn; jdliang@sjtu.edu.cn

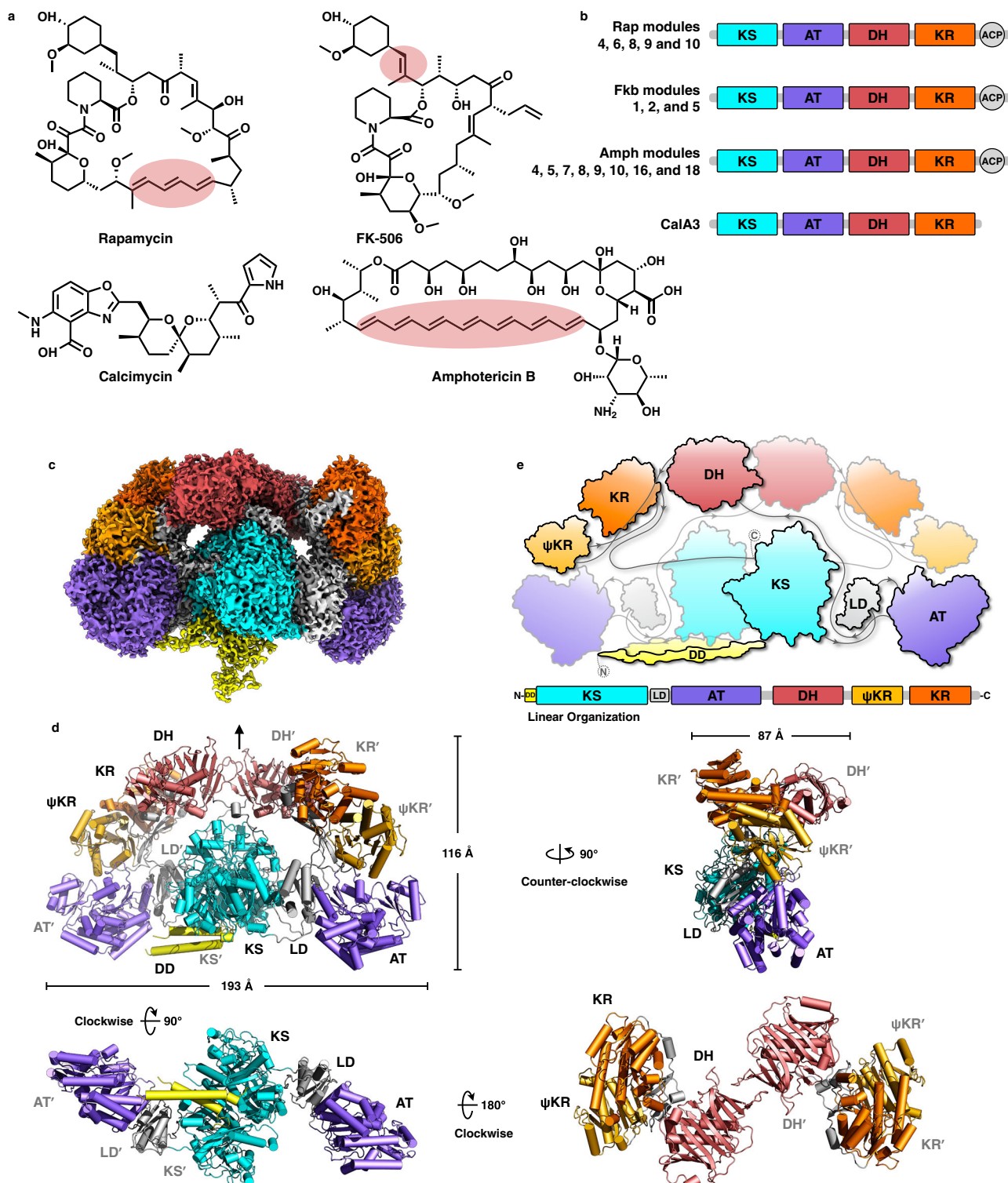

**Fig. 1 | Overall architecture of the dimeric CalA3 protein. a** Polyketide examples. Shared double bonds formed by DH domain dehydrations are highlighted with red color. **b** Domain organizations for KS-AT-DH-KR-ACP arranged PKSs. Five Rap modules, three Fkb modules, and eight Amph modules are involved in the biosynthesis of rapamycin (immunosuppressant), FK-506 (immunosuppressant), and amphotericin B (antibiotic), respectively. Domain arrangement-shared CalA3 is responsible for the biosynthesis of calcimycin. **c** Cryo-EM map of CalA3 at overall resolution of 3.38 Å. **d** Atomic model of CalA3 is shown in front, top, bottom and side views, and the dimensions are indicated. The pseudo-twofold symmetry axis is indicated by a black line with an arrow. The two protomers are differentiated by the apostrophe symbol. **e** Schematic diagram illustrating the "∞" shape of the domain arrangement of CalA3. The linear domain organization is shown at an approximate sequence scale, and each domain is colored with a unique color. Abbreviations used for each domain: DD N-terminal docking domain, LD linker domain, KS β-ketoacyl synthase, AT acyltransferase, DH dehydratase, KR ketoreductase, ψKR ketoreductase that contains only half of the typical active KR structure, ACP acyl carrier protein.

assembly and conformational changes. However, as for the keto-synthase (KS)-acyltransferase (AT)-dehydratase (DH)-ketoreductase (KR)-acyl carrier protein (ACP) (i.e., KS-AT-DH-KR-ACP) domain organized modular PKSs, it is not yet fully understood how these domains are assembled and interact with each other. Moreover, distinct from the typical C−C bond-forming condensation catalyzed by KS domains, how a KS releases a polyketide chain through aminoacylation has never been structurally characterized before.

In this work, we determined the structures of CalA3 using cryo-electron microscopy. CalA3 is a modular PKS involved in the biosynthesis of calcimycin[24]. The tightly compact structure results from the interactions between the catalytic and the structural region through LD-AT and ψKR/KR interfaces. The two catalytic chambers are symmetric, including the likely inactive domains of AT, DH, and KR, and the flexible DD. By in vitro reconstitution, molecular dynamics simulations and mutagenic analysis, we identified key residues capable of catalyzing amidation reaction. In comparison with AlphaFold2 predicted architectures of the modules mediating olefinic bond formation in rapamycin, amphotericin B, and FK-506 synthesis, the structure of CalA3 reveals a "king size lid" composed of LD-AT-DH-ψKR/KR domains.

## Results

### CalA3 is involved in polyketide chain release in calcimycin synthesis

In calcimycin biosynthesis, a long-standing question is how the C-N bond of benzoxazole ring is formed and polyketide chain is released. Calcimycin is synthesized by modular CalA1, A2, A5, and A4[12], in which the domain organization matches well with the carbon backbone of the pyrrole 2-carboxylic acid-primed polyketide chain (Supplementary Fig. 2a). This chain reacts with 3-hydroxyanthranilic acid (3-HAA) in a hitherto unknown mechanism to form C−N bond in the benzoxazole ring system and released.

Indeed, CalA3 was not on our radar initially. Starting from the characterization of calcimycin biosynthetic gene cluster by Wu et al. in 2011[12], we systematically investigate the cluster by performing multiple gene knockout experiments. Then, we in vitro reconstituted activities of several enzymes encoded by the gene cluster. For example: CalM is a N-methyltransferase that catalyze N-methyl modification on benzoxazole ring of calcimycin (Wu et al.[25]); CalR3 negatively controls calcimycin biosynthesis (Gou et al.[26]); CalC functions as an ATP-dependent CoA ligase that converts cezomycin to cezomycin-CoA to further calcimycin biosynthesis (Wu et al.[27]); and especially CalG, a type II thioesterase (TE) which was initially predicted to release the polyketide chain, turns out to be a dedicated TE that recycles overactivated acyls during calcimycin biosynthesis (Wu et al.[28]). Besides, there are no other genes encoding a TE domain or a putative hydrolase in the cluster to release the polyketide chain (Supplementary Fig. 1).

The phenotype of cezomycin and calcimycin elimination, proved by gene knockout experiments, could only be caused by disruptions of two genes: calU2 and calA3. However, our group previously showed that CalU2 was responsible for the spiroketal ring formation[29]. After ruling out the role of CalU2 and the above-mentioned CalG for the polyketide chain release, we were then promoted to switch our attention to the function of CalA3.

By gene knockout and in trans complementation experiments, we observed that CalA3 was necessary for the production of calcimycin (Supplementary Fig. 2b, c). We speculated that CalA3 releases the CalA4 ACP-tethered polyketide chain by catalyzing unique C−N bond formation in the presence of the substrate 3-HAA.

### Overall architecture of CalA3

To gain further insights into this enzyme, we determined the structure of CalA3 by cryo-EM analysis and reconstituted the in vitro amide bond formation reaction. We purified full-length His-tagged CalA3 using a nickel column, then a heparin column, and finally a size-exclusion column (Supplementary Fig. 3a–c). The substrates (SNAC-N1 and 3-HAA, discussed later) were added into the CalA3 solution for cryo-EM sample preparation. We collected cryo-EM images with a K3 Summit direct electron detector equipped on a Titan Krios electron microscope, and RELION was used for image processing[30–34]. Rounds of 2D and 3D classification for particle selection and refinements were performed. The cryo-EM maps of CalA3 at overall resolutions of 3.38 Å and 4.55 Å were ultimately reconstructed (Supplementary Figs. 3d–f and 4). Although the two substrates were not observed, the resolution of the maps enabled us to reliably assign the individual domains, dissect their linker junctions, and build atomic models of CalA3 (Supplementary Fig. 5).

The dimeric CalA3 adopts an "∞" shape in both front and back views, with dimensions of approximately $193 \times 87 \times 116$ Å³ (Fig. 1c–e). CalA3 is composed of two regions: the lower condensing (KS-AT) and the upper modifying (DH-ψKR/KR) regions. Since CalA3 KS does not catalyze Claisen condensation and DH-KR is likely inactive (discussed later), it is more reasonable here to describe them as catalytic (KS-AT) and structural (DH-ψKR/KR) regions. Starting from the catalytic region, the N-terminal docking domain (DD)-tethered KS and AT domains are connected by the 3α3β fold-linker domain (LD). Rather than the arch-shaped conformation in PikAIII[15], the catalytic region observed in CalA3 adopts an extended conformation similar to the known structures[17,19,20,22,35–38], whereas the LD-AT regions have slightly different rotational offsets (Supplementary Fig. 8). The catalytic region is then linked to the DH didomain (Supplementary Fig. 6), followed by the ψKR/KR domains pointing in opposite directions, constituting the structural region. The architecture of the structural region in CalA3 is impacted by two factors: the interdomain arrangement of the DH didomain and the orientation of ψKR/KR relative to the DH domain. The DH didomain arranges itself in a linear organization with an interdomain angle of 197°, similar to the modular PKSs[39–42] and MAS-like PKS[20] but different from the "V"-shaped DHs of mFAS[22] and LovB[19] (Supplementary Fig. 9). CalA3, mFAS, LovB, and MAS-like PKS are superimposed on the DH domain of one protomer, and the last three ψKR/KRs are lifted up approximately 50-70° relative to that of CalA3 (Supplementary Fig. 10a–d, f). Together, the modifying regions reveal the various relative orientations of DH-ψKR/KRs in the family of PKSs and mFAS, similar to a set of swing wings (Supplementary Fig. 10e).

The two subunits of CalA3 contact each other extensively with a buried surface of approximately 5687 Å², including both homophilic and heterophilic interactions that involve more than 100 residues per chain (Supplementary Fig. 7). The coiled-coil DD and KS domains together contribute to the largest dimerization interface of 2288 Å², while the two DH protomers dimerize via their N-terminal β-sheets with a 495 Å² interface. Intriguingly, distinct from the catalytic and structural regions being loosely connected, which was observed in mFAS, LovB, and MAS-like PKS (Supplementary Fig. 10), the structural region of CalA3 substantially contacts the catalytic region (Fig. 2a, b), resulting in interfaces that account for nearly 50% of the total buried surface area, with approximately 1421 Å² on each side, mainly through hydrogen bonding and ionic interactions between the Linker$_{DH-KR}$/ψKR and LD-AT domains of opposite chains (Supplementary Fig. 7). This almost inseparable interaction mode makes CalA3 a tightly compact core with nearly absolute symmetry.

### The nonenzymatic domains play structure-stabilizing roles

The N-terminal docking domain is responsible for communicating with the upstream module[43]. We observed two different rotational locations of DD in CalA3. The cryo-EM map is not accurate enough to model the DD domain, which is most likely due to its flexibility; however, the orientation of DD domain is crystal clear (Supplementary Fig. 11). One adopts the typical nearly perpendicular location, such as modular PKSs[36], protruding outward from the KS dimer and not contacting the

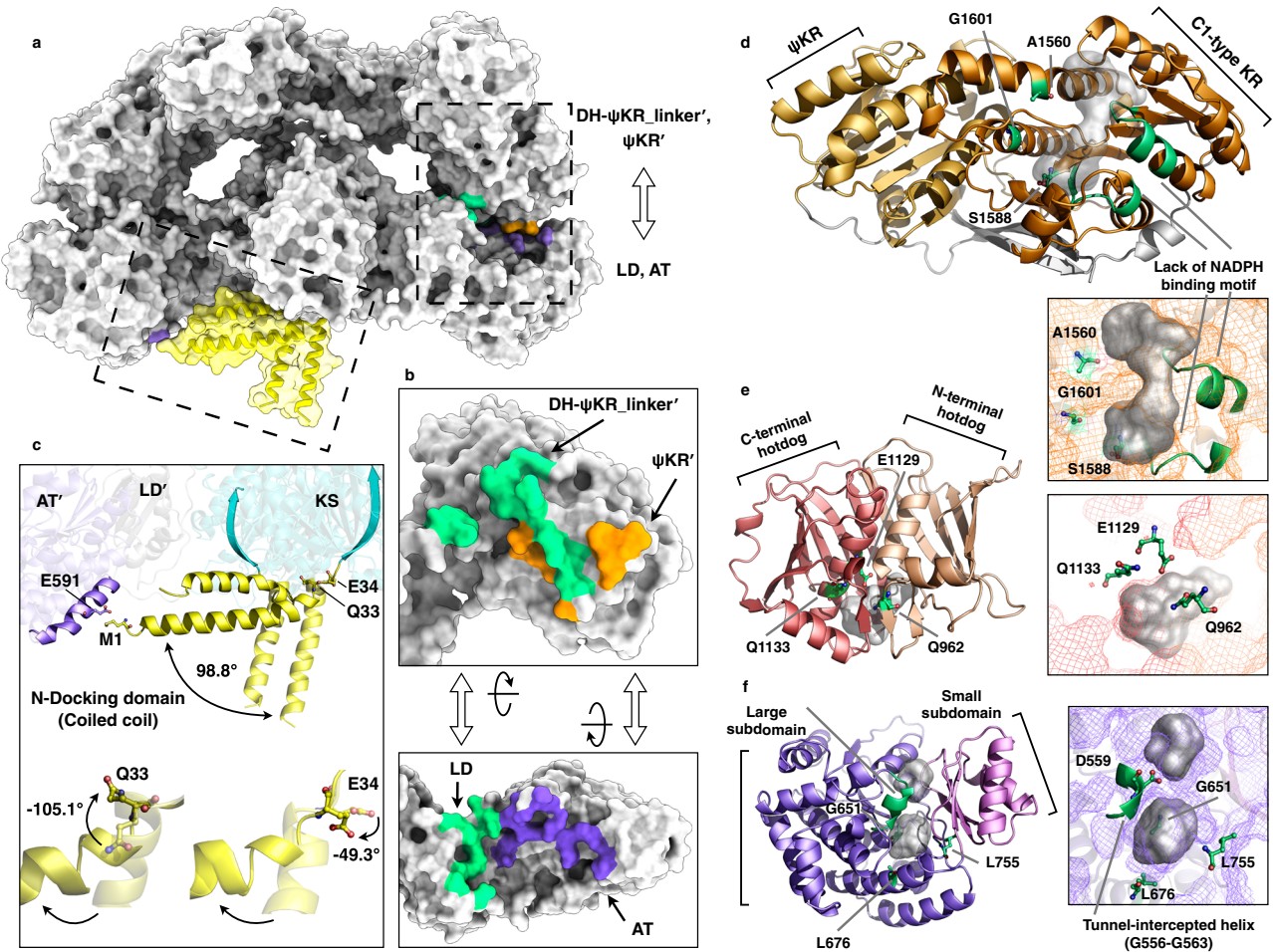

**Fig. 2 | Interdomain interactions and structural features of nonenzymatic domains. a** CalA3 atomic model is shown in gray surface. Bordered with dashed lines: contacts between ψKR/KR and LD-AT domains are shown in orange and purple, respectively; the interfaces contributed by LD or linker are colored green; two different locations of DD are shown in yellow. **b** ψKR/KR and LD-AT domains are split up from the whole structure, and rotated to the front views to highlight the interaction pattern. **c** Two distinct docking domain conformations. The nearly horizontal conformation of DD contacts with AT′ domain through two residues (shown as sticks). The rotational degree of DD and the residues that mediate this swing are labeled. **d** C1 type of ψKR/KR domain structure. **e** Two subdomain-arranged AT structure. **f** Two hotdog fold-adopted DH domain structure. These three inactive domains are both shown in ribbon representations, and the close-up views of active site regions are shown as mesh in (**e**), (**f**), and (**d**), respectively. The loss of canonical active site residues (shown as sticks) and the alteration of necessary structural elements are highlighted in green color. The cavities are shown in gray transparent surfaces. See also Supplementary Figs. 12–14 for more details.

rest of the protein. In contrast, the second conformation swings approximately 98.8° to the side, flanking the AT domain nearby through the contact between M1 and E591 (Fig. 2c). This rotation was mediated by residues Q33 and E34, with ψ dihedral angle-changes of −105.1° and −49.3°, respectively.

The AT domain is composed of a large α/β-hydrolase-like subdomain and a ferredoxin-like small subdomain (Fig. 2f), similar to the known AT and MAT structures[36,44–46]. Canonical ATs are responsible for selecting and loading extender units in polyketide biosynthesis[47]; however, it becomes an inactive version in CalA3 damaged by two structural features. The catalytic Ser-His dyad and the conserved Arg residue in homologs are altered to G651 and L755, and L676 in CalA3, respectively (Supplementary Fig. 12). A shifted loop truncates the substrate tunnel, which lies between the two subdomains (Fig. 2f and Supplementary Fig. 12).

The DH monomer adopts a pseudodimeric hot-dog fold (Fig. 2e). Despite the topological similarity with other DH domains[19,22,39,41], the mutation of active site residues harbored by both hot-dog structures makes it the second likely nonenzymatic domain of CalA3. Instead, the typical His, Asp and Gln (His) residues were changed to Q962, E1129 and Q1133, respectively (Fig. 2e and Supplementary Fig. 13). The cavity that is supposed to bind substrate between two hotdogs is more

structurally similar to mFAS and LovB but different from the traverse tunnel of modular PKSs (DEBS module 4 and CurK) with an open end (Supplementary Fig. 13).

The ψKR/KR domain contains an N-terminal structural subdomain (ψKR) and a C-terminal Rossmann-like subdomain, the latter of which observed in CalA3 is a unique C1-type KR (Fig. 2d). This C-terminal belongs to the short-chain dehydrogenase/reductase (SDR) superfamily[48] and shares a common structure with bacterial homologs. Nevertheless, distinct from all the available KR structures, including the A1, A2, B1, B2, and C2 types[49–53] (Supplementary Fig. 14), CalA3 KR completely loses its enzymatic function. The conserved Lys-Ser-Tyr catalytic triad is varied to A1560, S1588, and G1601. The NADPH-binding motif is altered, and the tunnel is shortened without the ability to bind nicotinamide coenzyme and the substrate (Supplementary Fig. 14). Considering the contact it makes with AT, CalA3 ψKR/KR only plays a structural role that stabilizes the whole structure.

In order to gain further insights into whether the LD-AT and DH-ψKR/KR are required for CalA3 function, we cloned, overexpressed, and tried to purify the two truncated versions of CalA3 (i.e., KS domain alone and KS-LD-AT tri-domain). Unfortunately, none of them have been successfully purified due to protein insolubility or failure of overexpression (Supplementary Fig. 15a, b). Furthermore, in order to

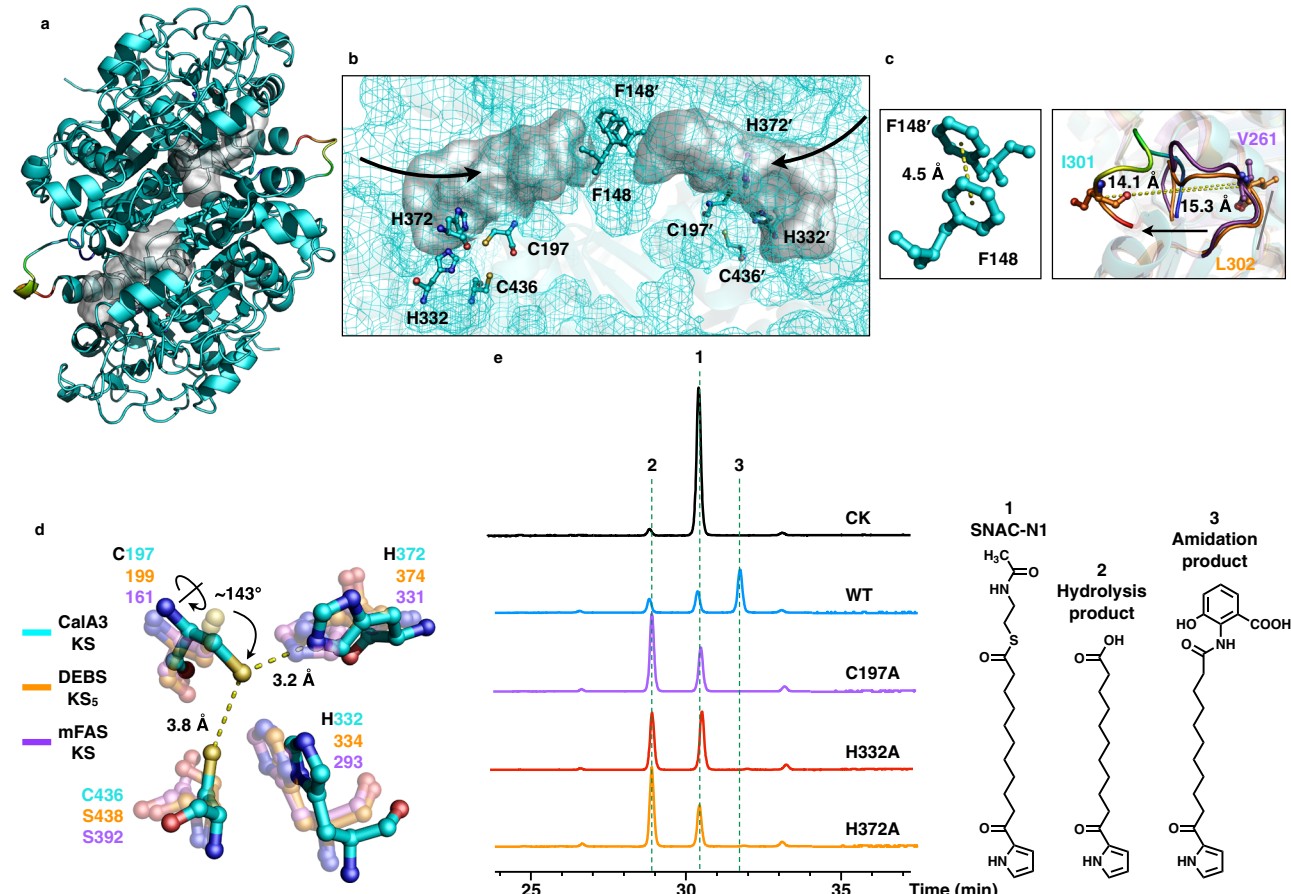

**Fig. 3 | Structure and mechanism of CalA3 KS domain. a** Structure of KS dimer, shown in ribbon representation. **b** Close-up view of the disconnected, double open-ended substrate tunnels. The atomic model of KS domain is shown here in mesh representation. Selected residues in (c-d) are shown as sticks. **c** Left, tunnel-intercepting aromatic residue pair. Right, superposition of CalA3 KS with DEBS module 5 and mFAS KS domains reveals a loop that shifts relatively 15.3 and 14.1 Å distance, respectively. **d** Superposition of the conserved Cys-His-His active site residues reveals a 143° side-chain rotation of C197 in CalA3 KS domain. The potential hydrogen bonds between C197-H372 and C197-C436 are indicated with yellow dashed lines and the distances are labeled (Å). **e** HPLC traces at $\lambda$ = 290 nm showing the products of the in vitro reactions catalyzed by CalA3 WT or mutants.

test whether excised AT and ψKR/KR domains would interact with each other (like the way the two domains interact in CalA3) and form a complex in vitro, we cloned AT and ψKR/KR domains out of CalA3 individually and tried to incubate the two domains in solution. However, there was only ψKR/KR domain that was successfully purified (Supplementary Fig. 15c, d). Together, we suggest that LD-AT-DH-ψKR/KR plays a structure-stabilizing role and are most likely required for CalA3 function.

**The KS domain catalyzes the hydrolysis and amidation reactions**

The ~420-residue KS domain contains the real catalytic function of CalA3. It possesses the typical thiolase fold containing an αβαβα structure[47] and closely resembles its homologs[22,36,54–56] (Supplementary Fig. 16). CalA3 KS has r.m.s deviations of 0.820 Å and 1.056 Å compared to DEBS module 5[36] and mFAS[22], respectively. Similarly, the substrate tunnel of the KS monomer traverses through with an open end (Supplementary Fig. 16); however, the two-sided tunnel of the KS dimer is intercepted by the F148 residue pair that forms the aromatic-aromatic interaction with a 4.5 Å distance (Fig. 3a–c). A loop (D294-S302) is conformationally shifted approximately 14.1 Å and 15.3 Å from DEBS module 5 and mFAS (Fig. 3c), respectively, and a corresponding loop was proven to undergo large movements during the dynamic gating mechanism in the elongating KS from *E. coli*[57]. The active site residues containing C197, H332, and H372 are conserved, but C197 rotates approximately 143° relative to the same positioned Cys residues in the

homologs (Fig. 3d and Supplementary Fig. 16). As a result, C197 is potentially capable of forming hydrogen bonds with both H372 and a lower C436 (Fig. 3d).

The canonical KS domain catalyzes chain extension through C–C bond formation, whereas it is clearly varied in CalA3. To ascertain the function of the KS domain and to examine the necessity for substrates, we synthesized SNAC-N1 (Fig. 3e, compound 1), the substrate analog, from 11-oxo-11-(pyrrol-2-yl) undecanoic acid and SNAC (Supplementary Fig. 28). SNAC-N1 and 3-HAA were incubated with CalA3, and a total of three peaks were detected by high-performance liquid chromatography (HPLC). Compared to the control (without the addition of CalA3), the dominant new product is compound 3 (Fig. 3e, trace WT), which forms by amidation between SNAC-N1 and 3-HAA, as confirmed by LC-high resolution mass spectrometry (LC-HRMS) and nuclear magnetic resonance (NMR) spectroscopy (Supplementary Figs. 19a–c, 29). The accumulation of 2, the hydrolysis product, was also observed. The hydrolysis of SNAC-N1 may be due to the open-ended tunnel of KS on each side, which is solvent-exposed. Indeed, the competition experiments indicate that the amidation reaction was increasingly favored over hydrolysis when the concentration of substrate 3-HAA was gradually raised (Supplementary Fig. 17).

To further investigate the amidation mechanism, we performed molecular dynamics (MD) simulations to analyze the interaction between SNAC-N1, 3-HAA and active site residues. The covalent SNAC-N1 spatially fitted the substrate tunnel very well. Due to the opposite

complementation of Coulombic electrostatic potential, 3-HAA is able to noncovalently bind at the substrate tunnel (Fig. 4a). Three independent simulations indicated that the amino group of 3-HAA was basically in the range (less than 4 Å, 29.8% proportion) of nucleophilic substitution to the C1 of C197-acylated SNAC-N1 in the prereaction state and remained stable for more than 50 ns (Supplementary Fig. 18a, b). Moreover, molecular mechanics/generalized born surface area (MM/GBSA) binding energy decomposition suggested that 3-HAA is stabilized by multiple interactions (Fig. 4c and Supplementary Fig. 18c). L439 may form hydrophobic interaction with the benzene ring of 3-HAA. The peptide bonds of L437 and L439 can form hydrogen bonds with the α-amino group of 3-HAA. H372 is at a distance of 2.9 Å and may undergo hydrogen bonding with 3-HAA, and the 3-HAA may be further reinforced by a salt bridge formed between the carboxyl group and the side chain lysyl of K367 (2.8 Å distance). A corresponding positioned lysine is verified to promote C-C bond formation in the DEBS module 1 KS domain[58], whereas it is suggested to stabilize the 3-HAA substrate in CalA3. The significance of the active site residues of the KS domain was also analyzed. Five mutants (C197A, H332A, H372A, L437A, and K367A) all completely abolished the amidation activities, and the hydrolysis products increased 4.46-, 3.38-, 4.15-, 4.86-, and 4.18-fold, respectively, compared to the WT (Supplementary Figs. 19d and 20). Together, holding 3-HAA in place by positioning residues is crucial for the amidation reaction.

To confirm the binding mode predicted by the MD simulations, we tried to prepare cryo-EM samples of CalA3 with the amidation product 3 and the hydrolysis product 2, respectively, and the two complex structures were finally obtained. The cryo-EM map of CalA3 with the amidation product was resolved at overall resolution of 3.84 Å and the amidation products were captured in the KS domains of both chain A and B (Fig. 4b and Supplementary Figs. 21 and 23). The benzoxazole moiety may undergo hydrophobic interactions with A336 and L439 and form additional hydrogen bond with R1718 which is at a distance of 3.5 Å (Fig. 4d). The polyketide chain and the pyrrole ring could also be positioned by hydrophobic interactions contributed by F236 and Q147 respectively. The cryo-EM map of CalA3 with the hydrolysis product was resolved at overall resolution of 3.97 Å and the hydrolysis product was captured only in the KS domain of chain A (Fig. 4b and Supplementary Figs. 22 and 23). The carboxyl oxygens may form hydrogen bonds with C197, T438, and L439 (Fig. 4e). Additionally, the carboxyl group of the hydrolysis product could also form a salt bridge with H372 at a distance of 5.4 Å which also demonstrates the positioning function of H372 predicted by MD simulations. Furthermore, we superposed the experimentally observed KS domain structures that contain the amidation and hydrolysis products respectively on the MD simulated KS domain structure that contains SNAC-N1 and 3-HAA to compare the relative locations of the substrates and the products (Fig. 4f). The C−N bond-formed benzoxazole moiety is shifted approximately 3.0 Å and rotated ~51.5° from the non-covalent 3-HAA substrate. The polyketide chain moieties are subtly moved within the range of 0.6 to 2.2 Å between the SNAC-N1, amidation, and hydrolysis products. The conformational variability of the pyrrole rings of the three compounds, however, is relatively significant. To consider the pyrrole and the carbonyl group of each chain as one unit, it is shifted ~6.5 Å and rotated ~53.6° from SNAC-N1 to the hydrolysis product, which the latter is able to further translate ~6.4 Å and rotate ~74.4° compared to the amidation product. Similarly, a motion of this unit combining movement of ~5.8 Å with rotation of 34.9° is also observed compared between the amidation product and SNAC-N1. Together, the visualization of the products in the KS domain verifies the basic binding location in the substrate tunnel analyzed by MD simulations and further reveals the structural flexibility of the polyketide chain.

Two pathways for the amidation mechanism were proposed (Supplementary Fig. 24a). By comparison of the energy barrier, amidation

pathway 1 with the assistance of H332 is 5.2 kcal/mol lower than pathway 2 where the amidation proceeds directly, and thus more energetically favored as 15.5 kcal/mol (Supplementary Fig. 24b). Based on the in vitro reconstitution, mutagenesis analysis, MD simulations and experimentally observed KS domain complex structures with products, we propose that the CalA4 ACP-tethered polyketide chain first enters the substrate tunnel of KS, and the nucleophile C197 attacks the thioester bond to be covalently acylated. H332 then may extract a hydrogen from the 3-HAA amino group, acting as a general base to perform the nucleophilic attack to eventually lead to amide formation or hydrolysis. During the formation of the carbon tetrahedral intermediate, H372, K367, L437, and L439 may perform the 3-HAA positioning function (Fig. 4g). The mechanism of polyketide chain release by C−N bond formation or thioester hydrolysis identified in this study indicates the catalytic versatility of the KS domain.

### Predictions and comparisons of the modular type I PKSs

Rather than the two loosely connected regions of currently reported PKSs and mFAS, CalA3 is extremely stabilized by complicated domain and linker interactions with almost exact symmetry. For fully functional PKSs with ACP that adopts the same architecture, the catalytic cycle may be hindered by steric constraints and mostly the lack of energetically coupled conformational dynamics (Fig. 5a). Directly opposite, the structural dynamics of modular PKSs are firmly established, featured by the end-to-end flip of ψKR/KR in PikAIII[16], the rotational ψKR/KR in Lsd14[17] and DEBS module 1[18], and the turnstile mechanism of AT in DEBS module 1[18]. They both illustrate the structural rearrangements during substrate shuttling by the ACP domains (Fig. 5b, c). Although the possibility cannot be ruled out currently, it is unlikely that the CalA3 architecture-identical modular PKSs (containing ACP) can perform such ACP-tethered substrate shuttling. Indeed, the two reaction chambers are symmetric, and three out of five domains of CalA3 are likely inactive.

To further investigate the fully functional KS-AT-DH-KR-ACP-arranged modular PKSs, we used AlphaFold2 to predict the structures of Rap modules 4, 6, 8, 9, and 10; Fkb modules 1, 2, and 5; and Amph modules 4, 5, 7, 8, 9, 10, 16, and 18. We superposed the condensing and modifying regions of these sixteen structures on the KS and DH dimers of CalA3, respectively (Fig. 5d, e and Figs. 25–27). The series of structures all exclude the interactions between the condensing and modifying regions. As exemplified by Rap M9 (Fig. 5d), the ACP domain can easily visit the docking site of DH and KR during the synthetic cycle. For docking to the KS and AT domains, although the shuttling back and forth of only the ACP domain does not seem to be the most favorable route for efficient substrate transfer, the changing in chamber symmetry may provide a better solution. Due to the flexible linker-connected architectures, at each catalytic stage, the condensing and modifying regions are capable of rotating horizontally and/or perpendicularly (Fig. 5f, g) relative to one another. The ψKR/KR domain may also perform positional translation relative to the DH (Supplementary Fig. 25) domain and fulfill the additional structural dynamics, which together create two asymmetric catalytic chambers forming polyketide products one at a time.

### Discussion

As a nature of PKSs that big molecules build small, CalA3 carries it to the extreme. The ~360 kDa megaenzyme uses only the KS domain to catalyze one C−N bond formation (or hydrolysis) and subsequently release the polyketide chain. The KS domain is the most highly conserved domain in either the architecture or active site residues[47]. Indeed, there are limited KS domains which have been reported to catalyze noncanonical reactions. NonJ and NonK are highly homologous to known KSs and they catalyze the formation of C−O bond in nonactin biosynthesis rather than the canonical C−C bond[59]. CerJ is also a KS homolog that catalyzes the C−O bond formation in

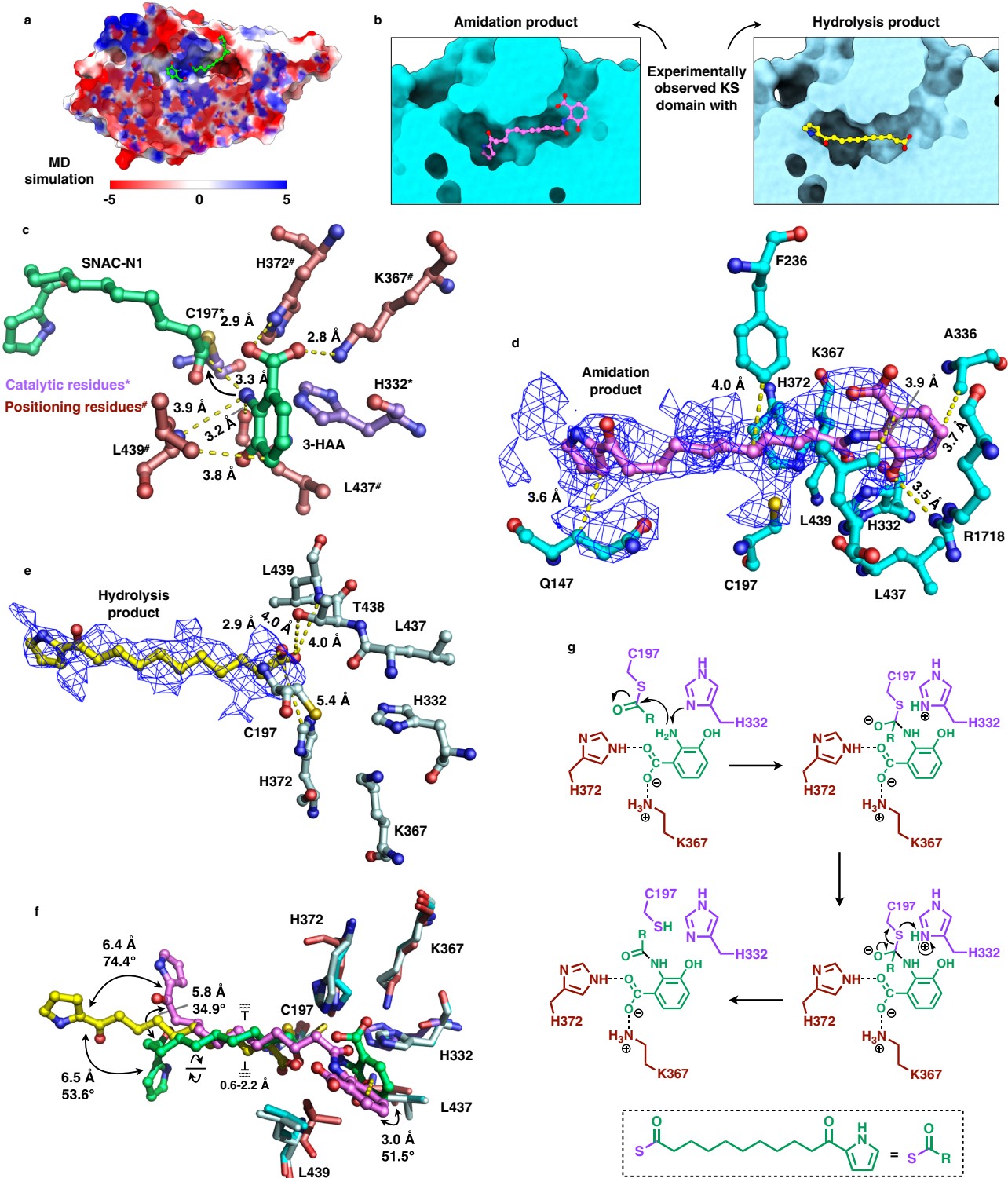

**Fig. 4 | Catalytic mechanism of CalA3 KS domain. a** Coulombic electrostatic potential (ESP) surface representation of KS domain, docked with 3-HAA and SNAC-N1 in the substrate tunnel. **b** Surface representation of KS domains captured with the amidation and hydrolysis products in the substrate tunnels respectively. **c** MD simulation of the covalently acylated polyketide chain (SNAC-N1) and the non-covalently docked 3-HAA in the active site region of KS domain. The substrates are coordinated by catalytic and positioning residues, which are marked with asterisk and hashtag, respectively. The distances between them are labeled (Å). Detailed views of the amidation product (**d**) and the hydrolysis product (**e**) binding in the active site region of KS domain, respectively. The cryo-EM map for each product is shown as mesh and carved at 2.5 Å distance. The map is contoured at 9.0σ for the amidation and at 7.6σ for the hydrolysis product, respectively. **f** Superposition of the products captured KS domains with MD simulated KS showing the conformational variability of each ligand. **g** Proposed amidation reaction mechanism.

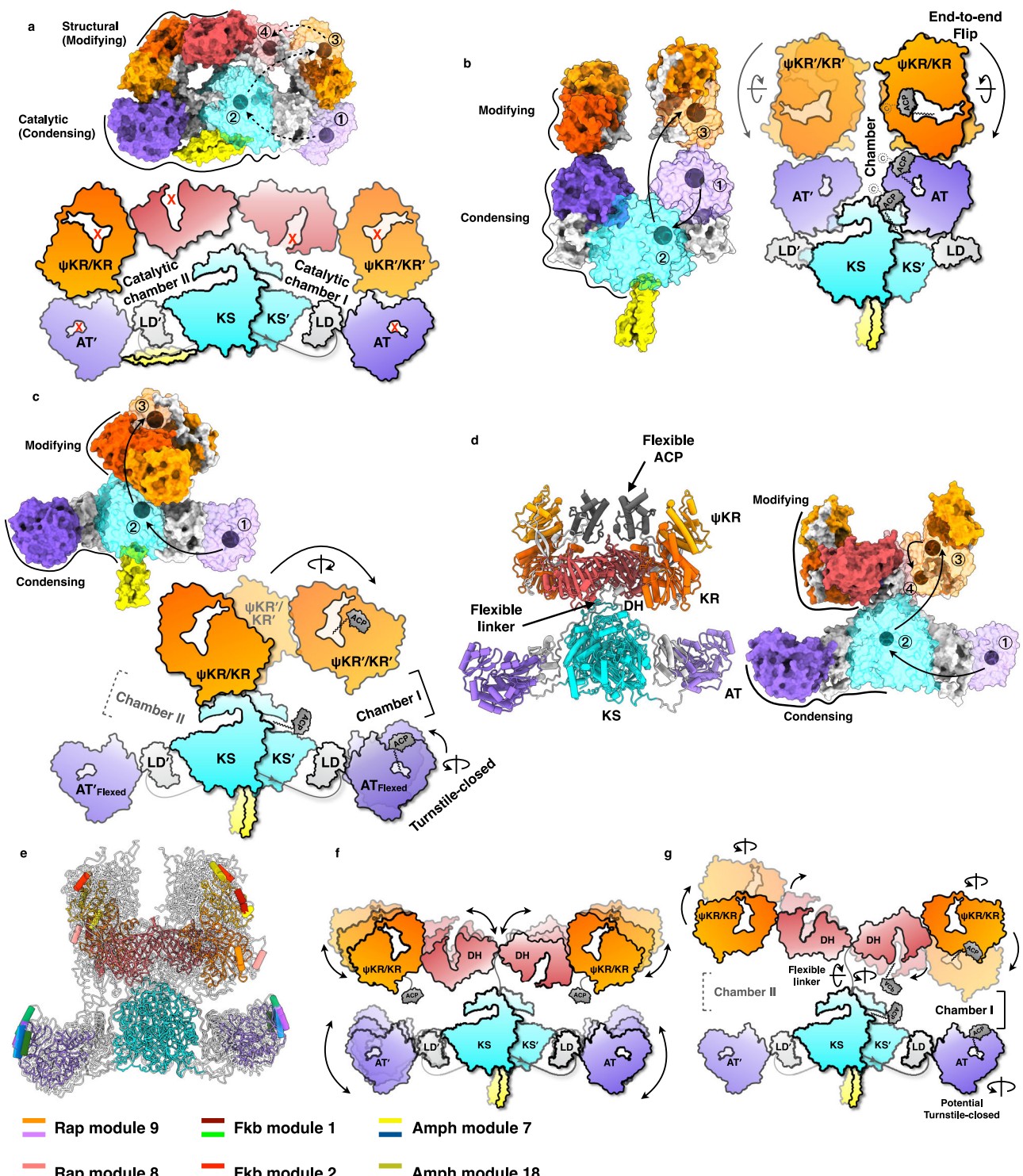

**Fig. 5 | Substrate shuttling within the catalytic chamber by the ACP domain.** The structures of CalA3, PikAIII[15], DEBS module 1[18], and AlphaFold2-predicted structures are shown as surface representations in (**a**–**d**), respectively. The hypothetical substrate shuttling trajectories (one side) within the catalytic chamber are shown as lines with each arrow pointing towards the next catalytic step; the active site residues locations of each domain are marked as gray balls. Schematic diagrams are depicted to illustrate the working mode of each modular PKS. **a** Steady CalA3 uses only KS domain to catalyze the polyketide chain release. **b** Structural dynamics of PikAIII. **c** Structural dynamics of Lsd14 and DEBS module 1. Note that the flexible AT domain is only observed in DEBS module 1. **d** A model of Rap module 9 structure shown in tubular and surface representations, generated by superposition of the condensing and modifying regions on the KS and DH dimers of CalA3, respectively. **e** Superposition of AlphaFold2-predicted structures of Rap modules 8, Fkb modules 1 and 2 and Amph modules 7 and 18 on the KS and DH domain dimers of Rap module 9. KS-AT-DH-KR domains of Rap M9 are colored cyan, purple, dark red, and orange, respectively. Other modular PKSs are transparent with only a helix of AT and a helix of KR domain shown in various colors to indicate the structural differences. The DDs and the flexible ACP domains are omitted on purpose. **f** Schematic diagrams illustrating the "X" shape of the domain arrangements of the predicted structures. **g** Potential structural dynamics of the KS-AT-DH-KR-ACP domain-organized PKSs during substrate shuttling process.

cervimycin biosynthesis[60]. KS domain in TAS1 which is a NRPS-PKS hybrid enzyme is involved in the cyclization step for TeA release[61]. KS domain in branching module of RhiE catalyzes vinylogous chain branching in rhizoxin biosynthesis[62,63]. In particular, the function of an unusual KS-like enzyme, BomK, was firmly established by Lv et al.[64]. BomK is responsible for amide bond formation in antibiotic A33853 biosynthesis, which provides the first clue for the KS-catalyzed C–N bond formation during the biosynthesis of benzoxazole-contained natural products. Nevertheless, by combining structural characterization using cryo-EM with biochemical characterization and MD simulations, the CalA3 KS-catalyzed C–N bond formation and chain release improved our understanding of functional divergence of apparent KS domains. Considering that the LD-AT-DH-ψKR/KR domains of CalA3 are likely nonenzymatic, CalA3 KS may also be perceived as a terminal amidase or hydroxylase, and functional CalA3 may be viewed as a gigantic type II enzyme for product release, which distinguished CalA3 from the above-mentioned PKSs. Moreover, the structure of CalA3 KS adopts the typical C–H–H active site residues, suggesting no major catalytic core reconfiguration. Given the similarity between the mechanism of KS and TE (both employing a Ser/Cys nucleophile and a basic residue), the repurposed KS domain of CalA3 reflects its own catalytic potential.

The surrounding domains tightly contacting each other around the KS are unusual. The architecture suggests a possibility that the entire LD-AT-DH-ψKR/KR may have been evolved into a "king size lid". These domains may lose canonical enzymatic function either by mutation or alteration during evolution[65] and play only a structural role. Similar case was reported in the curacin biosynthetic pathway[66]. The loading module CurA contains methyltransferase-like (MT_L), GCN5-related N-acetyltransferase-like (GNAT_L) and loading-ACP domains (MT_L-GNAT_L-ACP_L). However, the "MT_L" lost its ancestral function and was evolved into a lid-like subdomain, and the "GNAT_L" functions as a catalytic decarboxylase subdomain[67]. Thus, CurA "MT_L-GNAT_L" is, in fact, a "ψ decarboxylase/decarboxylase" domain. Nevertheless, structural rearrangements may be a requirement for fully functional PKSs, as exemplified by reported megaenzymes and predicted structures.

Our study uncovers an unusual architecture of PKS that contains the KS-AT-DH-KR domain arrangements and reveals the mechanism of the polyketide chain-releasing KS domain that can catalyze amidation or hydrolysis. The close structural comparisons between CalA3, the previously reported PKSs and FAS systems, and the AI-predicted structures illustrate the architecture and linkage between domains and provide insights into the rational re-engineering of chimeric modular PKSs to produce novel medicinal drugs.

## Methods

### Strains, plasmids, and culture conditions
The strains, plasmids, and primers used in this work are listed in Supplementary Table 1 and Supplementary Tabel 2. *Escherichia coli* plasmid isolation, gene cloning, and the molecular biological procedures were performed as described by Sambrook and Russell[68]. *S. chartreusis* NRRL 3882 genomic DNA was isolated according to the protocol of Kieser et al.[69].

Bacterial cells were routinely cultured in Luria broth medium (10 g/L tryptone, 5 g/L yeast extract, and 10 g/L sodium chloride) supplemented with 50 µg/mL kanamycin. Small-scale growth of *S. chartreusis* NRRL3882 and its derivative strains was by culture in tryptic soy yeast extract (TSBY) liquid medium, containing 3% tryptone soy broth, 10.3% sucrose, and 0.5% yeast extract (for extraction of chromosomal DNA), or on soya flour mannitol (SFM) agar containing 2% mannitol, 2% soybean powder, 2% agar (pH 7.2) (for sporulation and conjugation). Large-scale growth of *S. chartreusis* NRRL 3882 and its derivative mutant strains was performed in SFM medium without agar. Media were supplemented with 50 µg/ml apramycin when necessary.

### Cloning and expression of CalA3-cHis and its derived mutants
The *calA3* gene (5178 bp) was amplified by PCR using purified genomic DNA extracted from *S. chartreusis* NRRL 3882 and primers f28calA3cH-S and f28calA3cH-A (Supplementary Table 2), and plasmid vector bone was amplified by PCR using Plasmid pET28a (+) as template and primers v28calA3cH-S and v28calA3cH-A (Supplementary Table 2). The gene fragment and vector bone fragment (DpnI digested) of PCR amplification product, which were the ends of the two pairs of primers overlapped by 15–18 bp (Supplementary Table 2, base underlined and bold font) were fused by recombinant cloning kit (ClonExpress® Ultra One Step Cloning Kit by Vazyme biotechnology company). The resulting plasmid pET28CalA3_cH contains 6 × His Tags at the C-terminus with of CalA3 protein (180 kDa). For the construction of CalA3 mutants, plasmid pET28CalA3_cH was used as the template and primers with a partial target mutated base pair were listed in Supplementary Table 2 (base underlined and bold font). The amplified DNA fragments were digested using DpnI to remove the template DNA, and then transformed into *E. coli* DH10B. After confirmation by sequencing, pET28CalA3_cH and the derived mutational plasmids were transformed into *E. coli* BAP1 for protein production.

The *E. coli* strains used in protein expression experiments were grown in 1 L of LB medium containing 50 µg/ml kanamycin at 37 °C while shaking at 220 rpm to an OD of 0.6 at 600 nm, and then induced by the addition of isopropyl-_D_-thiogalactopyranoside (IPTG) to a final concentration of 0.1 mM. The culture was allowed to incubate for an additional 24 h at 16 °C. Cells were harvested by centrifugation at 6000 × *g* for 20 min, flash-frozen and stored at 4 °C for subsequent protein purification.

### Disruption and complementation of the *calA3* gene
The *calA3* gene in *S. chartreusis* was disrupted by inserting the apramycin resistance gene *aac*[12] IV using λ-Red technology. The *aac* IV-oriT cassette was amplified from the pIJ773 plasmid using primers calA3-F1 and calA3-F2 (Supplementary Table 2). The obtained PCR product (1.38 kb) was introduced into BW25113/pKD46 harboring the fosmid p16F9 targeting against the *calA3* gene, which generated the mutant plasmid pJTU3767 (ΔcalA3). The introduction of pJTU3767 into the *S. chartreusis* strains by conjugation and selection of double-crossover mutant strains through apramycin resistance were performed according to description by Kieser et al.[69]. Validation of the recombinant strain was performed by PCR analysis using primers calA3-F3 and calA3-F4.

For the complementation of strain GLX13 (ΔcalA3), the intact *calA3* gene was amplified from *S. chartreusis* NRRL 3882 genomic DNA with the primers calR3-F5 and calR3-F6 using a high-fidelity DNA polymerase (KOD-plus, TOYOBO). The resulting PCR fragment was cloned into an integrative plasmid, pJTU2170, which was derived from plasmid pIB139[70], resulting in pJTU3781. The *calA3*-complemented strain GLX14 (ΔcalA3::calA3) was generated by introducing pJTU3781 into GLX13 (ΔcalA3) via conjugation.

### Purification and sample preparations of His-tagged CalA3
Gravity columns were purchased from Sangon Biotech (Shanghai) Co., Ltd. TWEEN 20 and Millipore's Amicon® Ultra0.5 10k centrifugal filter devices were purchased from Sigma-Aldrich (Merck KGaA, Darmstadt, Germany). Ni-NTA Beads 6FF was purchased from Smart-Lifesciences (Smart-Lifesciences Biotechnology Co., Changzhou, China). HiTrap Heparin HP column and Superose 6 Increase 10/300 GL column were purchased from GE Healthcare (GE Healthcare Life Sciences, Little Chalfont, UK). All experiments were performed at 4 °C unless indicated.

For the purification of CalA3 wildtype and mutants, 20 g of frozen cells was thawed, resuspended in 100 mL of buffer A (50 mM HEPES pH 7.6, 20 mM imidazole, 150 mM NaCl, and 5% glycerol) and lysed using a French press (Union-Biotech, Shanghai, China) operated at 4 °C (600 bar). Cell debris was removed by centrifugation at 18,000 × *g* for 30 min, and the resulting supernatant was loaded onto a 24-mL gravity-

flow column packed with 4 mL of Ni-NTA Beads 6FF (Smart-Life-sciences) pre-equilibrated with 40 mL of buffer A. The resin was then washed with 40 mL of buffer B (50 mM HEPES pH 7.6, 20 mM imidazole, 750 mM NaCl, and 5% glycerol) to discard the unbound material. After the resin was re-washed with 20 ml buffer A to reduce the ionic strength, CalA3 was eluted using 25 mL of elution buffer (25 mM HEPES pH 7.6, 300 mM imidazole pH 7.6, 50 mM NaCl, and 5% glycerol). The resulting elution was applied to a 5 ml HiTrap Heparin HP column pre-equilibrated with 25 mL of buffer C (50 mM HEPES pH 7.6, 50 mM NaCl, and 5% glycerol), and further washed with 25 ml of buffer C. CalA3 was eluted with 10 ml of buffer D (50 mM HEPES pH 7.6, 500 mM NaCl, and 5% glycerol), concentrated to 1 mL using a Millipore Amicon® Ultra 10k centrifugal filter device according to the protocol provided by the company. The solution was centrifuged at $16,000 \times g$ for 10 min to remove precipitates and then subjected to size-exclusion chromatography using a pre-equilibrated Superose 6 Increase 10/300 GL column in sizing buffer (50 mM Tricine pH 8.0) on an ÄKTA fast protein liquid chromatography system (GE Healthcare Life Sciences). The peak fractions were pooled and concentrated with the centrifugal filter device to a concentration of approximately 4 mg/mL (determined by the Bradford assay using BSA as a standard). Approximately 6 mg of CalA3 can routinely be obtained from 20 g of cell paste. The sample was then combined with 3-HAA and SNAC-N1 at a final concentration of 1 mM for cryo-EM specimen preparation.

### In vitro reconstitution experiments

5 μM CalA3 WT or mutants were incubated with 0.5 mM 3HA and 0.15 mM SNAC-N1 in buffer (25 mM Tricine pH 7.5) in a 20 μL volume at 30 °C for 13 h. The reactions were quenched with 20 μL methanol. The precipitated protein was removed after centrifugation at $15,000 \times g$ for 20 min and the supernatant was prepared for HPLC and LC-MS analysis.

HPLC analysis was conducted with an Agilent C18 reverse-phase column (Pursuit XRs C18, 3 μm, 250 × 4.6 mm; Agilent) on an Agilent HPLC system (Agilent 1260 Infinity; Agilent) at a flow rate of 0.4 mL/min, a sample injection volume of 10 μL and mobile phase A (water/0.1% formic acid) and B (methanol/0.1% formic acid). Mobile A was gradually replaced by 75–85% volumes of mobile B over a period of 10 to 18 min, 85–95% volumes for 18–32 min, 95% volumes for 32–35 min. The elution was monitored by UV spectroscopy at $\lambda$ 290 nm and the amount of product present in the reaction mixtures was calculated from the peak area by Agilent OpenLAB CDS ChemStation Edition software.

LC-MS was conducted with an Agilent 1290 Infinity Liquid chromatography and 6545 Quadrupole Time-of-Flight Mass Spectrometer using positive electrospray ionization and an Agilent C18 reverse-phase column (ZORBAX StableBond C18, 5 μm, 250 × 4.6 mm; Agilent) on Agilent Masshunter Workstation software. The data are processed by Agilent Masshunter Qualitative Analysis Navigator. Samples were eluted at a flow rate of 0.3 mL/min with a sample injection volume of 10 μL and the same mobile phases as HPLC analysis. Mobile A was gradually replaced by 75–95% volumes of mobile B over a period of 8 to 22 min, 95% volumes for 22–30 min. Product masses was determined offline using ESI-MS in positive-ion mode and negative-ion mode with an ESI nebulizer and the mass spectrometer was set to acquire spectra in the mass range 50–1000 m/z.

### Synthesis of compound-S-(N-acetylcysteamine) substrate

Compound-S-(N-acetylcysteamine) substrates was synthesized using a published procedure with minor modifications. Briefly, a clean dry round-bottom flask (25 mL) with a magnetic stir bar was charged with compound 2 (2 mmol, 1 equiv), N-acetylcysteamine (0.24 g, 2 mmol, 1 equiv), DMAP (0.26 g, 2.2 mmol, 1.1 equiv), and EDC•HCl (0.42 g, 2.2 mmol, 1.1equiv). Dry dichloromethane (15 mL) was added and stirred at room temperature for 24 h. The reaction results were detected by HPLC. And the products were purified by C18 reversed-phase silica

gel (YMC, 50 μm) column, and eluted with a methanol/$H_2O$ gradient from 60 to 100%. The fractions from the reversed-phase silica gel column were analyzed with LC-MS. Fractions containing the target compounds were collected. The structures of the purified compounds were characterized using NMR. The NMR data were recorded on the Bruker NMR spectrometry (600 MHz). The NMR statistics are summarized in Supplementary Table 3 (SNAC-N1) and Supplementary Table 4 (amidation product).

### Competition experiment

Enzyme kinetic studies were performed for 3-HAA varying the substrate concentration in the range of 0–1000 μM in the presence of 11.7 nM CalA3. SNAC-N1 was supplied at 20 μM (due to the low solubility of SNAC-compound N1 in aqueous solution). The reaction was stopped after 10 min by adding isovolumetric ethyl acetate. The solvent of the recovered supernatant was removed by reduced pressure and the residue was resuspended in 60 μl methanol which was analyzed by HPLC. A gradient from 50-100% solvent B in 50 min (solvent A: 0.1% HCOOH in $H_2O$, solvent B: 100% $CH_3OH$) at a flow rate of 0.5 mL/min and a UV detection wave length of 290 nm was used.

### Cryo-EM specimen preparation and data acquisition

Four-microliter aliquots of specimens at ~4 mg/mL were applied to glow-discharged holey carbon grids (Quantifoil Cu, R1.2/1.3, 200 mesh) for 6 s of incubation, blotted for 2.5 s and plunge-frozen into liquid ethane precooled by liquid nitrogen using a Vitrobot Mark IV (FEI) operated at approximately 100% humidity and 4.5 °C. Cryo-EM images were collected with a Titan Krios electron microscope (FEI) operated at 300 kV and equipped with a K3 Summit direct electron detector (Gatan). Forty frames were recorded for each movie stack at a nominal magnification of 64000-fold in super-resolution mode with a pixel size of 1.1 Å within the defocus range of 1.5 to 2.5 μm. A total of 3299 movie stacks of CalA3 were automatically collected using EPU software with an exposure time of 2.16 s (0.1 s per frame) and a total dose of 48.6 e−/Å².

### Image processing

All movies of the dataset were aligned and dose-weighted using MotionCor2[30]. The contrast transfer function (CTF) and defocus parameters were determined by Gctf[31]. Micrograph checking, particle autopicking, 2D and 3D classification, autorefinement, postprocessing and resolution estimation of each cryo-EM map were performed using RELION 3.0[32–34]. Approximately 2000 particles of each dataset were manually picked and subjected to reference-free 2D classification. The best representative 2D classes were selected as templates for autopicking.

For the reconstruction of CalA3, the datasets were cleaned by removing ice contaminants and junk particles after two rounds of 2D classification, and good classes were kept to generate the 30-Å 3D initial model, which was low-pass filtered to 70 Å as the reference for subsequent 3D classification. The best classes of 224,991 particles were selected for autorefinement, which resulted in the reconstruction of a 3.72 Å CalA3 consensus cryo-EM map. Subsequent CTF refinement, Bayesian polishing, autorefinement and postprocessing were performed, yielding the 3.30 and 3.38 Å cryo-EM map imposed with C2 and C1 symmetry (Supplementary Fig. 3).

Three-dimensional classification with finer, local angular searches was further performed for conformational difference detection. For the nearly perpendicular conformation of docking domain, one class of 36,984 particles was observed, and further autorefined and postprocessed, which resulted in the reconstruction of a 4.55 Å cryo-EM map.

The resolution of all cryo-EM maps was estimated based on the corrected gold standard Fourier shell correlation (FSC) at the 0.143 criterion (Supplementary Fig. 4).

## Model building and refinement

First, the HHpred server was used for protein homology analysis using the HMM-HMM comparison method[71,72]. Multiple homologous crystal structures for each domain of CalA3 (DD, KS, AT, DH, and ψKR/KR) were rigid-body fitted into the cryo-EM maps using UCSF-Chimera[73] for comparative model rebuilding using RosettaCM[74–76]. The resulting atomic coordinates were further manually adjusted and built using Coot[77] and ISOLDE[78]. Structure refinement was performed using Phenix in real space with secondary structure and geometry restraints to prevent overfitting[79]. Subsequently, MolProbity[80] was used for model validation. The statistics are summarized in Supplementary Tables 5 and 6. All cryo-EM densities and atomic models were visualized, and the figures depicting them were prepared using PyMOL, UCSF-Chimera and ChimeraX[81].

## Cryo-EM specimen preparation, data acquisition, and image processing

Two sets of cryo-EM specimens were prepared: (1) CalA3+amidation product+3-HAA and (2) CalA3+hydrolysis product+3-HAA. For both sets, four-microliter aliquots of specimens at ~4.5 mg/mL were applied to glow-discharged holey carbon grids (C-Flat Au, R1.2/1.3, 300 mesh) for 6 s of incubation, blotted for 2.5 s and plunge-frozen into liquid ethane precooled by liquid nitrogen using a Vitrobot Mark IV (FEI) operated at approximately 100% humidity and 4.5 °C. Cryo-EM images were collected with a Titan Krios electron microscope (FEI) operated at 300 kV and equipped with a K3 Summit direct electron detector (Gatan). Forty frames were recorded for each movie stack at a nominal magnification of 81000-fold in super-resolution mode with a pixel size of 0.89 Å within the defocus range of 1.0–2.0 μm. A total of 2516 and 2280 movie stacks of CalA3 dataset (1) and (2), respectively, were automatically collected using EPU software with an exposure time of 2.67 s (0.1 s per frame) and a total dose of 48.06 e$^-$/Å$^2$.

The similar workflow was used to process both datasets separately. All movies of the datasets were aligned and dose-weighted using MotionCor2. The contrast transfer function (CTF) and defocus parameters were determined by CTFFIND 4[82]. Micrograph checking, particle autopicking, 2D and 3D classification, autorefinement, postprocessing and resolution estimation of each cryo-EM map were performed using RELION 4.0[83]. Approximately 2000 particles of each dataset were manually picked and subjected to reference-free 2D classification. The best representative 2D classes were selected as templates for autopicking.

For the reconstructions of CalA3, the datasets were cleaned by removing ice contaminants and junk particles after two rounds of 2D classification, and good classes were kept to generate the 30-Å 3D initial model, which was low-pass filtered to 70 Å as the reference for subsequent 3D classification. The best classes of 223,652 and 141,918 particles for (1) and (2) were selected for autorefinement, which resulted in the reconstructions of 4.16 and 4.38 Å CalA3 consensus cryo-EM maps, respectively. Subsequent CTF refinement, Bayesian polishing, autorefinement and postprocessing were performed, yielding the 3.84 and 3.97 Å cryo-EM map imposed with C1 symmetry (Supplementary Figs. 21, 22).

The resolutions of all cryo-EM maps were estimated based on the corrected gold standard Fourier shell correlation (FSC) at the 0.143 criterion (Supplementary Figs. 21f and 22f).

Model refinements and ligands building of the amidation and hydrolysis products were conducted using ISOLDE based on the CalA3 atomic model.

## Substrate tunnel generation

The substrate tunnel and cavities within each domain of CalA3 were calculated with the program Hollow[84] and adjusted using PyMOL (The PyMOL Molecular Graphics System, Version 2.2.3 Schrödinger, LLC.).

## Molecular dynamics (MD) simulations

The complex including the ketosynthase (KS), its covalent substrate SNAC-N1 and noncovalent 3-HAA, which was obtained by molecular docking via AutoDock[85], was prepared for MD simulations using Amber 18[86] program suite with ff14SB force field. The missing parameters for substrates were obtained from the RESP model where the atomic charges was calculated at the level of B3lyp/6-311 G(d,p) after structural optimizations at the level of B3lyp/6-31 G(d)[87]. All ionizable side chains were maintained in their protonation states at pH 6.68. The computational systems were immersed in an octahedral box of TIP3P water box and were neutralized with Na$^+$ using the AMBER18 LEAP module. Then proper minimizations were carried out to remove atomic collisions. After heating from 0 to 300 K in 50 ps, the systems were equilibrated for 50 ps to obtain a reasonable initial structure. Three 100 ns trajectories were obtained for each model based on the equilibrated structure.

## Binding free energy calculations

The molecular mechanics generalized Born surface area (MM-GBSA) method was used to calculate the binding free energy between 3-HAA and KS with a python program *MMPBSA.py*[88]. 50 snapshots extracted from 10 to 40 ns in one of trajectories were used to calculate the binding free energies. The decomposition of the energies was utilized to identify crucial residues in contributions to the interactions between 3-HAA and KS.

## Quantum mechanical/molecular mechanical (QM/MM) calculations

In order to study the catalytic mechanism of the KS domain, QM/MM calculations were carried out via Gaussian 09 program[89]. The initial structure was originated from the representative structures of MD simulations. The computational model consisted of truncated covalent substrate, 3-HAA, and truncated key side chains of residue H332, K367, and H372, 76 atoms in total. At the level of B3lyp/6–31 G(d), the optimizations of minimums, transition states, intermediates, and intrinsic reaction coordinate (IRC) were calculated. The transition states were confirmed by a single imaginary frequency and the correct vibrational vector. After that, B3lyp/6–311+G(d,p) level was used to calculate the single point energy with SMD[90] solvation model to improve the computational accuracy.

## AlphaFold2 predictions of the modular type I PKSs

ColabFold[91] was locally employed to predict Rap modules 4, 6, 8, 9, and 10, Fkb modules 1, 2, and 5 and Amph modules 4, 5, 7, 8, 9, 10, 16, and 18. Briefly, the condensing region containing the KS and AT domains and the modifying region containing the DH and KR domains of the modules were predicted separately. The parameters were set as homooligomer 2, max_recycle 3, use_ptm and use_turbo. A total of five models were generated with the top1 relaxed. The obtained models were then superposed to CalA3, resulting the final full model of the corresponding modules.

## Reporting summary

Further information on research design is available in the Nature Portfolio Reporting Summary linked to this article.

# Data availability

The 3D cryo-EM maps generated in this study have been deposited in the Electron Microscopy Data Bank [https://www.emdataresource.org/] under the accession numbers EMD-32863, EMD-32864, EMD-35188, and EMD-35189, (Supplementary Tables 5 and 6). The atomic coordinate has been deposited in the Protein Data Bank [https://www.rcsb.org] under the accession numbers 7WVZ, 8I4Y, and 8I4Z. Previously published structures cited in this study can be accessed using PDB accession numbers 2QO3, 2HG4, 4MZ0, 7S6B, 5BP1, 3HHD, 2VZ8,

7CPX, 3EL6, 4LN9, 3KG9, 5IOK, 5BP4, 1NM2, 3SBM, 4AMP, 3MJS, 4L4X, 2Z5L, 2FR1, 3QP9, 4NA2, 6KXF and 6SMP. Other data are available from the corresponding authors upon request. Source data are provided with this paper.

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

## Acknowledgements

We thank F. F. Wang, G. Y. Li, L. H. Xin, J. L. Duan, and N. Liu for their help with the sample preparation and data collection. Cryo-EM images were collected at the National Facility for Protein Science in Shanghai (NFPS), Zhangjiang Lab. The computations in this paper were run on the π 2.0 cluster supported by the Center for High Performance Computing at Shanghai Jiao Tong University. This work was financially supported by the National Key R&D Program of China (2019YFA0905400 and 2021YFA0910500 to J.L., 2018YFA0900700 to Z.W.), the National Science Foundation of China (32270033 to Z.W., 32201035 to J.W., 32271302 to J.L., 32150013 and 32171252 to Z.Deng), the China Postdoctoral Science Foundation (2022T150412 and 2022M720085 to J.W.) and Shanghai 'Super Postdoctoral' Incentive Program (202112 to J.W.).

## Author contributions

Z.W., J.W., and J.L. conceived the study. J.W. performed Cryo-EM analysis, built atomic models of protein structure, did structural analysis, prepared the figures, and wrote the paper; X.W., X.L., and D.L. characterized the biochemical activity of this protein; L.K. received cryo-EM data and particle optimization; Z.Du and T.S. finished the molecular dynamics simulations; L.G. manipulated genetic experiments on knockout and complementation; H.W., W.C., D.L. manipulated experiments on protein expression construction and purification process establishment; X.W. and S.L. analyzed the nuclear magnetic resonance data of the compounds; Z.W., J.W., and J.L. analyzed the data. All authors wrote the paper. J.L., Z.W., and Z.Deng supervised this project.

## Competing interests

The authors declare no competing interests.
