## [Peer Review File · Nature Communications]

C-N bond formation by a polyketide synthaseREVIEWER COMMENTS

Reviewer #1 (Remarks to the Author):

Modular PKS structures are especially attractive because of their incredible structural complexity and flexibility. In the recent years, development of cryo-EM techniques have empowered progress on structural determination of type I PKSs, such as modular PikAIII, Lsd14, DEBS module 1, as well as iterative MAS-like PKS, and LovB reported by the authors of this manuscript. However, no high-resolution structure of DH-KR or DH-ER-KR within a modular PKS has been reported, presenting a crucial knowledge gap in how the reductive loop is structurally organized in modular PKSs. In this manuscript, the authors determined a cryo-EM structure of CalA3, a "modular" PKS with a DD-KS-LD-AT-DH-KR domain composition. The structural analysis, comparison with known PKS module and domain structures, molecular dynamics simulation, together with in vivo and in vitro assays convincingly proved that CalA3 serves as the offloading/termination enzyme and utilizes its KS to catalyze amidation or hydrolysis. Other domains in CalA3 are nonfunctional and merely serve as structural domains. The authors also presented several other AlphaFold-predicted modular PKS structures with KS-AT-DH-KR-ACP composition for comparison.

Overall, the CalA3 structure is well described and illustrated in detail. However, in the biosynthetic pathway, ACP is absent in CalA3, and no carbon chain extension reaction occurs on CalA3 (3-HAA is not ACP-attached, so condensation of 3-HAA and SNAC substrate is not a typical modular PKS-catalyzed chain extension). The "modifying" DH-KR domains closely contact the KS-AT core/catalytic domains, which is in large contrast to known KR loosely-attached modular PKS structures (PikAIII, Lsd14 and DEBS module 1). Another major difference is that Lsd14 and DEBS module 1 modifying regions are both significantly asymmetric, and the asymmetry is likely important for modular PKS function, whereas CalA3 is pseudo-twofold symmetric. These key differences suggest that CalA3 cannot be viewed as a typical PKS module, and functionally CalA3 is more like a gigantic type II enzyme for product release. Thus, it is difficult to evaluate the general impact of the structural features unraveled by this study, especially how DH-KR is packed in PKS. In fact, the observation that CalA3 DH-KR tightly interacts with the core domains suggests a possibility that the entire DH-KR may have been evolved into a "king size lid". Similar cases were reported in other PKS systems. For example, in the curacin pathway, the loading module CurA contains MT-GNAT-ACP. The "MT" lost its ancestral function and was evolved into a lid-like subdomain and the "GNAT" is a catalytic decarboxylase subdomain. Thus, CurA "MT-GNAT" is in fact a " ψ decarboxylase/decarboxylase" domain. The authors need to present more data on whether the DH-KR is required for CalA3 function. In Extended Data Fig. 1c, Δ calA3::calA3_KS did not restore the product formation. One explanation is that removing LD-AT destabilized the KS and disabled its function. The authors should present the result with Δ calA3::calA3_KS-LD-AT as well, which will indicate whether DH-KR presence is required for CalA3 function. Given the fact that modular PKSs often contain inactive domains, the major novelty for this study is that CalA3 KS serves as a terminal amidase or hydroxylase. Functional divergence of apparent KS domains in different modular PKSs/hybrid PKSs is widely recognized, KSQ and KS0 for example. Moreover, CalA3 KS adopts the same catalytic triads, suggesting no major catalytic core reconfiguration. In sum, despite of the nice work that the authors carried out on CalA3, I am doubtful that this study will significantly improve our understanding of modular PKSs or benefit rational re-engineering of chimeric modular PKSs.

Other comments:

Page 2 Line 59-60: "The two symmetric catalytic chambers are dysfunctional,..." This is an inaccurate statement, since no evidence rules out the possibility that the cognate ACP sits in one of the two chambers to allow KS catalyzing product release. If that is the case, then the chambers are in fact functional, except that none of the canonical AT-, DH-, KR-catalyzed reactions would happen there.

Page 2 Line 73: Please state the full chemical name of 3-HAA here.

Page 3 Line 87: CalA3 is composed of two regions: the lower condensing (KS-AT) and the upper modifying (DH- ψ KR/KR) regions. I think it is more reasonable to describe the

lower region as the catalytic region and the upper as the structural region in this manuscript, since CalA3 KS does not catalyze Claisen condensation and CalA3 DH-KR is inactive.

Page 3 Line 104: "The two monomers..." should be "The two subunits".

Page 4 Line 155: "A loop is conformationally shifted.." Please mention the residue numbers of the loop in CalA3.

Figure 1d: It is hard to see whether the 90o or 180o rotation axis is clockwise or counter-clockwise.

Figure 2: It may be better to show DH structure in 2e, and AT structure in panel 2f, since AT is below DH-KR in 2a.

Figure 3e: It is recommended to add two more mutant (C436A, C436S) assay results here to test the role of C436 in catalysis. It is intriguing that C436 is close to the catalytic C197. 3.8 angstroms distance is a bit far for disulfide bond formation, but given the 3.4 angstroms resolution of this structure, I wonder what is the distance variation between C197 and C436 sulfurs during molecular dynamics simulation when the acylated SNAC-N1 is removed.

Extended Data Fig. 1c: The arrow illustration doesn't clearly show that which peak (the 39 min one or the 40 min one) represents Cezomycin. It is recommended to use a dashed line to highlight each product peak here.

Extended Data Fig. 19: It is hard to evaluate the necessity of this data for the whole study, since CalA3 DH-KR is nonfunctional, and without a experimentally determined reference structure, the credibility of predicted structures of large flexible complexes is questionable. For better assessment of these PKS model structures, it is recommended to present a predicted alignment error (PAE) plot which indicates AlphaFold's confidence in domain packing, as well as contact probability plot (If two domains or modules have a low relative contact probability, they are more likely to interact.).

In Extended Data Table 1, please list the detailed information of Δ calA3::calA3_KS.

Reviewer #2 (Remarks to the Author):

Overall, this is an interesting study that combines structural characterization using cryoEM with biochemical characterization and MD simulations for an unusual PKS.

1. It should be mentioned in the abstract that CalA3's function is chain release, to not confuse the reader with canonical PKS modules that perform chain elongation and beta-group modification. The meaning of the descriptor used in the abstract, "dysfunctional", is not clear (Line 31).

2. In line 122, authors wrote "Although there is no apparent evidence, the swingable N-terminal docking domain may play a role in finding the upstream module (CalA4) and subsequent interaction." Please expand on this. How would this swinging mechanism aid finding the upstream module?

3. In Fig. 4, it is difficult to tell apart panels c and d. Please rearrange.

4. Authors claim in line 247 "Our study uncovers a basic model of PKSs that contain the KS-AT-DH-KR domain arrangements" This statement seems to suggest that the current structure is representative of PKS modules containing the KS-AT-DH-KR set of domains. However, AlphaFold results provided by the authors suggests the opposite. Please rephrase or clarify.

5. Please specify which maps are shown in Figure 3b and, Extended data figures 4 and 5 – are these unsharpened maps or sharpened maps?

6. Map density in Fig. 3b is unclear. There is no map density corresponding to residue C197. In this case, the authors' claim regarding C197 conformation in lines 158-161 is not supported by the experimental map density.

7. Line 94 - Map density for the AT-DH linker shown in Extended Data figure 5 is unclear. In this case, it is not clear if the map density in this region is accurate enough to model the linker. Similarly, the map density for the DD domain in Extended data figure 4 seems unclear.

8. Please provide map threshold values for all the maps shown in the manuscript.

9. Line 95 - "The architecture of modification" should be "The architecture of the modifying region".

10. To increase the readability of the manuscript for a general scientific audience, the authors should improve clarity in writing. For example, in line 108, the authors should specify what they are referring to by "two regions". This should be "the modifying and the condensing regions".

11. In addition to the interaction between the DH-KR linker and the LD-AT domains, does the AT-DH linker also contribute to bringing the condensing and the modifying regions close in CalA3? Is the AT-DH linker in CalA3 shorter than that in other modular PKSs, MAS-like PKS and mFAS? Or does the linker retain the length but attains a different conformation compared to that observed in the MAS-like PKS and mFAS structures thereby bringing the condensing and the modifying regions closer?

12. Lines 132-134 - The proposed inactivation of the DH domain has not been confirmed using biochemical characterization. It is possible that Q962, E1129 and Q1133 can still maintain DH catalytic activity as the mutations are not vastly different from the typical residues: His, Asp and Gln (His). I suggest writing "likely nonenzymatic" or "likely inactive" instead of "nonenzymatic".

13. Since the cryoEM sample contained SNAC-N1 and 3-HAA, did the authors observe any map density for these compounds in the KS active site? Please also specify whether the map density of SNAC-N1 or 3-HAA was observed in the main text.

14. Please add labels for compounds corresponding to the peaks as SNAC-N1, hydrolysis product and amidation product in Fig. 3e and Extended data fig. 4.

15. Have other PKSs having a domain organization like CalA3 (KS-AT-DH-KR and no ACP) been identified in other PKS pathways? Are the proposed inactivating features of the AT, DH, KR domains and the linker interactions that contribute to the rigid symmetric architecture of CalA3 conserved in any other PKSs?

Reviewer #3 (Remarks to the Author):

This manuscript reports the biochemical and structural characterization of an unusual polyketide synthase (PKS) module, CalA3, which is responsible for the amide bond formation in the biosynthesis of calcimycin. Initially, the authors revealed the involvement of CalA3 in the biosynthesis by gene deletion experiment, then obtained the cryo-EM structure, and finally performed in vitro enzymatic reactions, providing insight into how the unusual amide bond formation is catalyzed by CalA3. I found that the enzymatic reaction reported herein and the detailed structural analysis of CalA3 are of general interest; however, I would like to raise several concerns, which should be addressed before further consideration of the manuscript.

1. I think that the Introduction section does not provide sufficient information for readers. Provide more information about calcimycin biosynthesis and explain why the biosynthesis is intriguing and should be studied. Also, it is not very clear why the authors suddenly mention the polyketide chain release by aminoacylation in lines 54-55.

Some other minor issues are: (i) simvastatin is not a pure natural product (line 39); (ii) abbreviations, such as AT and DH, should be defined when they appear for the first time in the manuscript.

2. The authors initially revealed that CalA3 is engaged in the biosynthesis by gene deletion experiment. This subsection is very briefly written, and it is not very clear why the authors focused only on CalA3 out of the enzymes encoded by the gene cluster. Also, the gene deletion experiment resulted in the elimination of calcimycin, but no accumulation of another metabolite was observed. Based on this result, I feel that it is difficult to predict that CalA3 is the enzyme for the polyketide chain release. Please provide a more reasonable explanation. In addition, please provide an image of the whole gene cluster in the supporting information.

3. Related to the previous comment, it is not very clear to me why the authors first solved the structure of CalA3 before understanding the catalytic function of this enzyme.

4. The authors mention that the noncatalytic domains contribute to the stabilization of the protein structure. However, this hypothesis was not experimentally investigated. The authors should perform in vitro enzymatic reactions using the standalone KS domain and some other truncated versions of CalA3 to further understand the roles of the noncatalytic domains.

5. CalA3 can hydrolyze compound 1 to form 2, and the authors state that the hydrolysis might be "due to the open-ended tunnel of KS" (lines 171-172). The catalytic mechanism of this hydrolysis is not very clear, although it seems the catalytic residues of the KS are not used for the reaction. Please provide a more detailed explanation for this chemical reaction.

6. This manuscript lacks a discussion on the comparison of the CalA3-catalyzed reaction and other noncanonical reactions catalyzed by KS-like enzymes. Especially, the authors' group previously discovered KS-catalyzed amide bond formation in the biosynthesis of A33853 (Chem. Biol., 2015, 22, 1313-1324), but this work is not cited in this manuscript. Please emphasize the novelty of the CalA3-catalyzed reaction in comparison with similar preceding cases.

Other minor points:

1. Line 164 – Does SNAC-N1 correspond to compound 1? Please clarify.
2. Fig. 3f – Some arrows and charges are missing in the figure.
3. References are not provided in a unified manner.
4. Extended Data Fig. 1 – "H20" should be a typo of "H2O".

Point-by-point response to the reviewers

*Reviewer Comments: Black, Helvetica, 10

*Author Responses: Blue, Helvetica, 10

Note: "A." stands for "Answer:".

Reviewer #1 (Remarks to the Author):

Modular PKS structures are especially attractive because of their incredible structural complexity and flexibility. In the recent years, development of cryo-EM techniques have empowered progress on structural determination of type I PKSs, such as modular PikAIII, Lsd14, DEBS module 1, as well as iterative MAS-like PKS, and LovB reported by the authors of this manuscript. However, no high-resolution structure of DH-KR or DH-ER-KR within a modular PKS has been reported, presenting a crucial knowledge gap in how the reductive loop is structurally organized in modular PKSs. In this manuscript, the authors determined a cryo-EM structure of CalA3, a "modular" PKS with a DD-KS-LD-AT-DH-KR domain composition. The structural analysis, comparison with known PKS module and domain structures, molecular dynamics simulation, together with in vivo and in vitro assays convincingly proved that CalA3 serves as the offloading/termination enzyme and utilizes its KS to catalyze amidation or hydrolysis. Other domains in CalA3 are nonfunctional and merely serve as structural domains. The authors also presented several other AlphaFold-predicted modular PKS structures with KS-AT-DH-KR-ACP composition for comparison.

Overall, the CalA3 structure is well described and illustrated in detail. However, in the biosynthetic pathway, ACP is absent in CalA3, and no carbon chain extension reaction occurs on CalA3 (3-HAA is not ACP-attached, so condensation of 3-HAA and SNAC substrate is not a typical modular PKS-catalyzed chain extension). The "modifying" DH-KR domains closely contact the KS-AT core/catalytic domains, which is in large contrast to known KR loosely-attached modular PKS structures (PikAIII, Lsd14 and DEBS module 1). Another major difference is that Lsd14 and DEBS module 1 modifying regions are both significantly asymmetric, and the asymmetry is likely important for modular PKS function, whereas CalA3 is pseudo-twofold symmetric. These key differences suggest that CalA3 cannot be viewed as a typical PKS module, and functionally CalA3 is more like a gigantic type II enzyme for product release. Thus, it is difficult to evaluate the general impact of the structural features unraveled by this study, especially how DH-KR is packed in PKS. In fact, the observation that CalA3 DH-KR tightly interacts with the core domains suggests a possibility that the entire DH-KR may have been evolved into a "king size lid". Similar cases were reported in other PKS systems. For example, in the curacin pathway, the loading module CurA contains MT-GNAT-ACP. The "MT" lost its ancestral function and was evolved into a lid-like subdomain and the "GNAT" is a catalytic decarboxylase subdomain. Thus, CurA "MT-GNAT" is in fact a "ψ decarboxylase/decarboxylase" domain. The authors need to present more data on whether the DH-KR is required for CalA3 function. In Extended Data Fig. 1c, Δ calA3::calA3_KS did not restore the product formation. One explanation is that removing LD-AT destabilized the KS and disabled its function. The authors should present the result with Δ calA3::calA3_KS-LD-AT as well, which will indicate whether DH-KR presence is required for CalA3 function. Given the fact that modular PKSs often contain inactive domains, the major novelty for this study is that CalA3 KS serves as a terminal amidase or hydroxylase. Functional divergence of apparent KS domains in different modular PKSs/hybrid PKSs is widely recognized, KSQ and KS0 for example. Moreover, CalA3 KS adopts the same catalytic triads, suggesting no major catalytic core reconfiguration. In sum, despite of the nice work that the authors carried out on CalA3, I am doubtful that this study will significantly improve our understanding of modular PKSs or benefit rational re-engineering of chimeric modular PKSs.

We highly appreciate the reviewer's insightful and critical comments on our work. These comments are extremely professional and very helpful to the quality promotion of this manuscript. We have carefully addressed these comments point by point and rewritten/reorganized the manuscript for more clarity at our best. New truncation, mutation and MD simulation experiments have been performed for better supporting our conclusion, as the reviewer suggested.

A: We would like to response to the major concerns from three points.

1. We agree that CalA3 cannot be viewed as a typical modular PKS (absence of ACP, not a typical modular PKS-catalyzed chain extension reaction and pseudo-twofold symmetry). From this point of view, we have revised the title of the manuscript as “C-N bond formation by a polyketide synthase”. We have also clearly mentioned the CalA3’s chain release function in the abstract (Line 25): “Here, we present the cryo-EM structure of CalA3, a chain release PKS module without an APC domain, at 3.38 Å resolution.”.

2. Good point! The reviewer’s perspective of the “king size lid” is unique and quite insightful. We have now added this point in the discussion section.

Line 278: “The surrounding domains tightly contacting each other around the KS are unusual. The architecture suggests a possibility that the entire LD-AT-DH- ψ KR/KR may have been evolved into a “king size lid”. These domains may lose canonical enzymatic function either by mutation or alteration during evolution and play only a structural stabilizing role. Similar case was reported in the curacin biosynthetic pathway¹. The loading module CurA contains methyltransferase-like (MT_L), GCN5-related N-acetyltransferase-like (GNAT_L) and loading-ACP domains (MT_L-GNAT_L-ACPL). However, the “MT_L” lost its ancestral function and was evolved into a lid-like subdomain and the “GNAT_L” functions as a catalytic decarboxylase subdomain². Thus, CurA “MT_L-GNAT_L” is in fact a “ ψ decarboxylase/decarboxylase” domain. Nevertheless, structural rearrangements may be a requirement for fully functional PKSs, as exemplified by reported megaenzymes and predicted structures.”

1. Gu, L. et al. GNAT-like strategy for polyketide chain initiation. *Science* **318**, 970–974 (2007).

2. Skiba, M. A. et al. Repurposing the GNAT fold in the initiation of polyketide biosynthesis. *Structure* **28**, 63-74 (2020).

3. In order to gain further insights into whether the LD-AT and DH- ψ KR/KR are required for CalA3 function, we cloned, overexpressed and tried to purify the two truncated versions of CalA3 (i.e., KS domain alone and KS-LD-AT tri-domain). Unfortunately, none of them have been successfully purified due to protein insolubility or failure of overexpression (Figure_Response 1a-b). Furthermore, in order to test whether excised AT and ψ KR/KR domains would interact with each other (like the way the two domains interact in CalA3) and form a complex *in vitro*, we cloned AT and ψ KR/KR domains out of CalA3 individually and tried to incubate the two domains in solution. However, there was only ψ KR/KR domain that was successfully purified (Figure_Response 1c-d).

The CalA3 truncation experiments indicated that the deletion of either LD-AT-DH- ψ KR/KR domains or DH- ψ KR/KR domains destabilized the KS and disabled its function, hence we supposed that the complementation of gene knockout ($\Delta calA3::calA3_{KS-LD-AT}$) would unlikely to restore the *in vivo* product formation.

So yes, we agree with the reviewer’s opinion and suggest that LD-AT-DH- ψ KR/KR plays a structure-stabilizing role and are most likely required for CalA3 function.

Figure_Response 1. SDS-PAGE analysis of truncated versions of CalA3. The targeted theoretical molecular weight of each truncation is labeled in **a**, KS domain; **b**, KS-LD-AT domain; **c**, AT domain; **d**, ψ KR/KR domain. Note that only ψ KR/KR domain is successfully purified. Abbreviations used for each gel: M, marker; T, total protein; S, supernatant; Ft, flow through; E_Ni, elution of His-tagged protein; E_Strep, elution of StrepII-tagged protein, SEC, size exclusion chromatography.

Figure_Response 1a-d are currently not incorporated in the manuscript (unless required to do so by the reviewer #1).

Other comments:

Page 2 Line 59-60: “The two symmetric catalytic chambers are dysfunctional,...” This is an inaccurate statement, since no evidence rules out the possibility that the cognate ACP sits in one of the two chambers to allow KS catalyzing product release. If that is the case, then the chambers are in fact functional, except that none of the canonical AT-, DH-, KR-catalyzed reactions would happen there.

A: Thanks! We agree that this sentence is imprecise. ACP of CalA4 (upstream module) would likely to sit in chambers to deliver the polyketide chain to allow KS of CalA3 catalyzing product release.

We have deleted the “dysfunctional” and revised this sentence and similar descriptions throughout the text as follows (Line 69): “The two catalytic chambers are symmetric....”

Page 2 Line 73: Please state the full chemical name of 3-HAA here.

A: We have now stated the full chemical name of 3-HAA when it appear for the first time in the manuscript. Line 79: “This chain reacts with 3-hydroxyanthranilic acid (3-HAA) in a hitherto unknown mechanism...”

Page 3 Line 87: CalA3 is composed of two regions: the lower condensing (KS-AT) and the upper modifying (DH-ψKR/KR) regions. I think it is more reasonable to describe the lower region as the catalytic region and the upper as the structural region in this manuscript, since CalA3 KS does not catalyze Claisen condensation and CalA3 DH-KR is inactive.

A: We fully agree with the reviewer’s point on the terminology of the two regions of CalA3. For not causing confusion in the PKSs field, we mentioned the traditional “condensing/modifying” only once and used “catalytic/structural regions” in the rest of the manuscript.

The sentence has been revised as follows (Line 102): “CalA3 is composed of two regions: the lower condensing (KS-AT) and the upper modifying (DH-ψKR/KR) regions. Since CalA3 KS does not catalyze Claisen condensation and DH-KR is likely inactive (discussed later), it is more reasonable here to describe them as catalytic (KS-AT) and structural (DH-ψKR/KR) regions. Starting from the catalytic region.....”

Page 3 Line 104: “The two monomers...” should be “The two subunits”.

A: It has been revised as “The two subunits of CalA3 contact each other extensively...” (Line 121).

Page 4 Line 155: “A loop is conformationally shifted..” Please mention the residue numbers of the loop in CalA3.

A: The residue numbers have been added. Line 172: “A loop (D294-S302) is conformationally shifted...”

Figure 1d: It is hard to see whether the 90o or 180o rotation axis is clockwise or counter-clockwise.

A: We have now clearly labeled the “clockwise or counter-clockwise” next to the rotation axes.

Figure 2: It may be better to show DH structure in 2e, and AT structure in panel 2f, since AT is below DH-KR in 2a.

A: Fully agree. We have now changed the display orders of DH and AT structures in Fig. 2.

Figure 3e: It is recommended to add two more mutant (C436A, C436S) assay results here to test the role of C436 in catalysis. It is intriguing that C436 is close to the catalytic C197. 3.8 angstroms distance is a bit far for disulfide bond formation, but given the 3.4 angstroms resolution of this structure, I wonder what is the distance variation between C197 and C436 sulfurs during molecular dynamics simulation when the acylated SNAC-N1 is removed.

A: In order to test the role of C436 in catalysis and its relative position variation, we performed the mutation assay of C436A and molecular dynamics simulation when the acylated SNAC-N1 is removed.

HPLC analyses showed that neither the activity of amidation nor hydrolysis was abolished, which indicates that the C436A mutant did not significantly affect the activities of CalA3 (Figure_Response 2a-b).

Three parallel MD simulations without any substrate (SNAC-N1 or 3-HAA) showed that the distance between C197 and C436 sulfur atoms varied from 4-10 Å and the average distance for MD1, MD2 and MD3 was 5.8 Å, 6.8 Å and 6.1 Å, respectively. Because the length of a disulfide bond is approximately 2.05 Å, the distance distributions of Sulfur_{C197}-Sulfur_{C436} suggest that they will not form a disulfide bond.

Based on the experiments of mutation and MD simulations, we conclude that C436 is unlikely to be involved in catalysis of CalA3.

Figure_Response 2. Analysis of the role of CalA3 C436 in catalysis. **a**, SDS-PAGE analysis of purified CalA3 C436A mutant. **b**, HPLC traces at $\lambda=290$ nm showing the products of the *in vitro* reactions catalyzed by CalA3 WT or C436A mutant. **c**, The distance variation between C197 and C436 sulfurs over time of 100 ns in three parallel MD simulations (MD1-MD3). **d**, The distance between C197 and C436 in the initial and the most dominant cluster from each MD trajectory (MD1-MD3).

Figure_Response 2a-d are currently not incorporated in the manuscript (unless required to do so by the reviewer #1).

Extended Data Fig. 1c: The arrow illustration doesn't clearly show that which peak (the 39 min one or the 40 min one) represents Cezomycin. It is recommended to use a dashed line to highlight each product peak here.

A: Sorry for the unclarity and the 40 min HPLC peak represents cezomycin here. We have now used a dashed line to highlight each product peak.

Extended Data Fig. 19: It is hard to evaluate the necessity of this data for the whole study, since CalA3 DH-KR is nonfunctional, and without an experimentally determined reference structure, the credibility of predicted structures of large flexible complexes is questionable. For better assessment of these PKS model structures, it is recommended to present a predicted alignment error (PAE) plot which indicates AlphaFold's confidence in domain packing, as well as contact probability plot (If two domains or modules have a low relative contact probability, they are more likely to interact.).

A: We agree that the AlphaFold2-predicted structures lack an experimentally determined reference structure. We have now provided the PAE plot and contact probability plot of each predicted module (Figure_Response 3-4, i.e., Supplementary Figs. 21-22).

Figure_Response 3 (Supplementary Fig. 21). The confidence prediction of the AlphaFold2-predicted structures (part 1). Each panel of Amph M4, 5, 7, 8, 9, 10, 16 and 18 shows two plots. Left, the predicted aligned error (PAE) plot which helps to assess the reliability of relative domain positions and orientations; right, the contact probability plot which indicates the probability of the two domains interaction.

Figure_Response 4 (Supplementary Fig. 21). The confidence prediction of the AlphaFold2-predicted structures (part 2). Each panel of Fkb M1, M2 and M5; Rap M4, M6, M8, M9 and M10 shows two plots. Left, the predicted aligned error (PAE) plot which helps to assess the reliability of relative domain positions and orientations; right, the contact probability plot which indicates the probability of the two domains interaction.

In Extended Data Table 1, please list the detailed information of $\Delta calA3::calA3_{KS}$.

A: Sorry for the careless. We have now listed the detailed information of $\Delta calA3::calA3_{KS}$ in Supplementary Table 1.

Supplementary Table 1. Bacterial strains and plasmids used in this study.
(only the $\Delta calA3::calA3_{KS}$ -related information is shown in this response document).

Strain or plasmid	Description	Reference or source
GLX15 ($\Delta calA3::calA3_KS$)	Complementation strain of the calA3-KS gene fragment of Streptomyces chartreusis NRRL 3882 disrupted calA3 gene mutant	This work
Plasmids		
pJTU3782	pJTU2170-derived plasmid carrying calA3-KS for expression in Streptomyces chartreusis (PCR amplification fragment of calA3-KS (1356 bp)), NdeI and EcoRI were introduced into the two ends of the PCR fragment, which was connected to the corresponding sites of pJTU2170 and used to complement the calA3 -deletion mutant	This work

Reviewer #2 (Remarks to the Author):

Overall, this is an interesting study that combines structural characterization using cryoEM with biochemical characterization and MD simulations for an unusual PKS.

We are thankful for your highly positive comments on our work. These comments are very helpful to the quality promotion of this manuscript. The points have been carefully addressed point by point, the new figures have been added and revised for better supporting our conclusion. The manuscript has been rephrased and clarified to a clearer understandable version for readers, thanks to your professional suggestions.

1. It should be mentioned in the abstract that CalA3's function is chain release, to not confuse the reader with canonical PKS modules that perform chain elongation and beta-group modification. The meaning of the descriptor used in the abstract, "dysfunctional", is not clear (Line 31).

A: We agree that CalA3 should not be viewed as a canonical modular PKS (1. absence of ACP; 2. not a typical modular PKS which catalyzes Claisen condensation and beta-group modification; 3. pseudo-twofold symmetry). We have now mentioned the CalA3's chain release function in the abstract (Line 25): "Here, we present the cryo-EM structure of CalA3, a chain release PKS module without an APC domain, at 3.38 Å resolution.". We have also revised the title of the manuscript as "C-N bond formation by a polyketide synthase".

We agree that the descriptor of "dysfunctional" used in the abstract is imprecise. We presume that ACP of CalA4 (upstream module) would likely to sit in chambers to deliver the polyketide chain to allow KS of CalA3 catalyzing polyketide chain release.

We have deleted the "dysfunctional" in the abstract and revised it and the similar descriptions throughout the text. Line 29: ".....resulting in two stabilized chambers with nearly perfect symmetry."

2. In line 122, authors wrote "Although there is no apparent evidence, the swingable N-terminal docking domain may play a role in finding the upstream module (CalA4) and subsequent interaction." Please expand on this. How would this swinging mechanism aid finding the upstream module?

A: We think that this sentence is overstated. We have deleted this description because of there is currently no further experimental evidence supporting this swinging phenomenon.

3. In Fig. 4, it is difficult to tell apart panels c and d. Please rearrange.

A: Sorry for the unclarity. We have now switched the display orders of panels c and d in Fig. 4.

4. Authors claim in line 247 “Our study uncovers a basic model of PKSs that contain the KS-AT-DH-KR domain arrangements” This statement seems to suggest that the current structure is representative of PKS modules containing the KS-AT-DH-KR set of domains. However, AlphaFold results provided by the authors suggests the opposite. Please rephrase or clarify.

A: We agree. Because CalA3 should not be viewed as a canonical modular PKS, we have rephrased this sentence as follows (Line 289): “Our study uncovers an unusual architecture of PKS that contains the KS-AT-DH-KR domain arrangements and...”.

5. Please specify which maps are shown in Figure 3b and, Extended data figures 4 and 5 – are these unsharpened maps or sharpened maps?

A: It is unsharpened map shown in Extended data Fig. 4 (Supplementary Fig. 5) and sharpened map shown in Extended data Fig. 5 (Supplementary Fig. 6). We have now stated in each figure legend whether the map shown is sharpened or unsharpened.

6. Map density in Fig. 3b is unclear. There is no map density corresponding to residue C197. In this case, the authors' claim regarding C197 conformation in lines 158-161 is not supported by the experimental map density.

A: Sorry for this confusion. The Fig. 3b is shown in mesh representation of the atomic model, not the density map and we have now clearly stated this in the legend of Fig. 3b. We have now also shown the map density of active site residues (C197-H372-H332) in Supplementary Fig. 5c.

Figure_Response 1 (Supplementary Fig. 5). c, The DeepEMhancer¹-enhanced cryo-EM map for each chain of the active site residues of KS domain. The carving distance is 3 Å. The map is contoured at 0.035 (0.9 σ) for chain A and 0.085 (2.1 σ) for chain B.

1. Sanchez-Garcia, R. et al. DeepEMhancer: a deep learning solution for cryo-EM volume post-processing. *Commun. Biol.* **4**, 874 (2021).

7. Line 94 - Map density for the AT-DH linker shown in Extended Data figure 5 is unclear. In this case, it is not clear if the map density in this region is accurate enough to model the linker. Similarly, the map density for the DD domain in Extended data figure 4 seems unclear.

A: For better integrity of the AT-DH linker map density, we have now used the unsharpened map in Extended Data Fig. 5 (Supplementary Fig. 6).

Figure_Response 2 (Supplementary Fig. 6). Linker-based domain organization of CalA3.....

To improve the map interpretability of DD domain, we have used DeepEMhancer¹ software to post-process (sharpen-like) the CalA3 cryo-EM maps. The orientation of DD domain is crystal clear; however, the map density in this case is not accurate enough to model the DD domain which is most likely due to its flexibility. We have now deleted the panel of full-atom represented DD atomic model-fitted density map and clearly stated in the figure legend that the DD domain is only docked.

Figure_Response 3. The cryo-EM maps of CalA3 DD domain. Left, unsharpened map contoured at 0.039 (1.6 σ); right, DeepEMhancer sharpened map contoured at 0.003 (0.07 σ).

1. Sanchez-Garcia, R. et al. DeepEMhancer: a deep learning solution for cryo-EM volume post-processing. *Commun. Biol.* 4, 874 (2021).

Figure_Response 3 is currently not incorporated in the manuscript (unless required to do so by the reviewer #2).

8. Please provide map threshold values for all the maps shown in the manuscript.

A: The map threshold values for all the maps have been provided in each figure legend.

9. Line 95 – “The architecture of modification” should be “The architecture of the modifying region”.

A: Thanks, the sentence has been revised.

Besides (related to reviewer’s point 1), since CalA3 should not be viewed as a canonical modular PKS and its KS does not catalyze Claisen condensation and DH-KR is likely inactive, we have rephrased the terminology of the two regions of CalA3 as catalytic and structural regions. For not causing confusion in the PKSs field, we mentioned the traditional “condensing/modifying” of canonical PKSs only once and used “catalytic/structural regions” for CalA3 in the rest of the manuscript.

For example, Line 102: “CalA3 is composed of two regions: the lower condensing (KS-AT) and the upper modifying (DH- ψ KR/KR) regions. Since CalA3 KS does not catalyze Claisen condensation and DH-KR is likely inactive (discussed later), it is more reasonable here to describe them as catalytic (KS-AT) and structural (DH- ψ KR/KR) regions. Starting from the catalytic region.....”

10. To increase the readability of the manuscript for a general scientific audience, the authors should improve clarity in writing. For example, in line 108, the authors should specify what they are referring to by “two regions”. This should be “the modifying and the condensing regions”.

A: We have revised this sentence as follows: “Intriguingly, distinct from the catalytic and structural regions being loosely connected...” (Line 125).

To increase the readability for a general scientific audience, we have also provided more information about calcimycin biosynthesis and its biochemical interest in the introduction section.

Line 48: “The polyether antibiotic calcimycin (A23187) in *Streptomyces chartreusis* NRRL 3882 is widely used as a biochemical tool in pharmacological and *in vitro* toxicological studies¹. As a divalent cation ionophore², calcimycin is also able to uncouple oxidative phosphorylation of mammalian cells³, inhibit ATPase activity⁴ and induce apoptosis via activation of intracellular signaling⁵. Structurally, calcimycin is composed of an α -ketopyrrole, a spiroketal ring (i.e., a polyketide) which is synthesized by the above-mentioned type I modular PKSs, and a substituted benzoxazole moiety which involves an unusual C-N bond formation and product release⁶ (Supplementary Fig. 1). Reported by X-ray crystallographic studies^{7,8}, the C-N bond contained benzoxazole system is crucial for calcium ion binding and thus indispensable for this ionophore-mediated biological processes. However, the mechanism by which this unique amidation reaction and chain release in calcimycin biosynthesis remains unknown.”

1. H., B. J. & A., V. H. J. The use of the divalent calcium-ionophore A23187 as a biochemical tool in pharmacological and *in vitro* toxicological studies. *Cell Struct. Funct.* **21**, 97–99 (1996).

2. Pressman, B. C. Biological applications of ionophores. *Annu. Rev. Biochem.* **45**, 501–530 (1976).

3. Andreo, C. S. & Vallejos, R. H. Uncoupling of photophosphorylation in spinach chloroplasts by the ionophorous antibiotic A23187. *FEBS Lett.* **46**, 343–346 (1974).

4. Hara, H. & Kanazawa, T. Selective inhibition by ionophore A23187 of the enzyme isomerization in the catalytic cycle of sarcoplasmic reticulum Ca^{2+} -ATPase. *J. Biol. Chem.* **261**, 16584–16590 (1986).

5. Kajitani, N. et al. Mechanism of A23187-induced apoptosis in HL-60 cells: dependency on mitochondrial permeability transition but not on NADPH oxidase. *Biosci. Biotechnol. Biochem.* **71**, 2701–2711 (2007).

6. Wu, Q. et al. Characterization of the biosynthesis gene cluster for the pyrrole polyether antibiotic calcimycin (A23187) in *Streptomyces chartreusis* NRRL 3882. *Antimicrob. Agents Chemother.* **55**, 974–982 (2011).

7. Chaney, M. O., Jones, N. D. & Debono, M. The structure of the calcium complex of A23187, a divalent cation ionophore antibiotic. *J. Antibiot.* **29**, 424–427 (1976).

8. Smith, G. D. & Duax, W. L. Crystal and molecular structure of the calcium ion complex of A23187. *J. Am. Chem. Soc.* **98**, 1578–1580 (1976).

11. In addition to the interaction between the DH-KR linker and the LD-AT domains, does the AT-DH linker also contribute to bringing the condensing and the modifying regions close in CalA3? Is the AT-DH linker in CalA3 shorter than that in other modular PKSs, MAS-like PKS and mFAS? Or does the linker retain the length but attains a different conformation compared to that observed in the MAS-like PKS and mFAS structures thereby bringing the condensing and the modifying regions closer?

A: This is an interesting question. In order to compare the linkers, we superposed the mFAS and LovB structures on the KS domain of CalA3 (Figure_Response 4a). The AT-DH linker of CalA3 (Gly897-Gly934), mFAS (Pro818-Ser857) and LovB (Asp915-Pro965) has 38, 40 and 51 amino acids, respectively and each linker adopts the relatively different conformation at the C-terminus of the linker. From a perspective of the DH domain position, the extent of the “closeness” between the two regions of three structures is fairly similar. The alignments showed that the condensing and the modifying regions being close does not necessarily lead to the interaction between the two regions. It seems that AT-DH linker is able to adopt similar length but result in the totally different architecture.

Furthermore, we superposed Alphafold2-predicted PpsD (Phenolphthiocerol/phthiocerol polyketide synthase subunit D) structure on the KS domain of CalA3 and then on the DH domain of CalA3 (Figure_Response 4b-c). The PpsD structure is more similar to CalA3. PpsD contains functional KS-MAT-DH-KR-ACP domain organization, but the modifying region does not contact the condensing region which

makes the comparison more obvious. The AT-DH linker of CalA3 (Gly897-Gly934) and PpsD (Thr878-Gly914) has 38 and 37 amino acids, respectively; the DH-ψKR/KR linker of CalA3 (Arg1194-Leu1209) and PpsD (Arg1194-Ile1205) has 16 and 12 amino acids, respectively. The alignments indicated that the length of the linkers may not be the decisive role, whereas the linkers may retain the similar length but attain a different conformation which finally leads to the domain positions with huge difference.

For now, it is hard to clearly evaluate the contribution of the different conformations of the AT-DH and the DH-ψKR/KR linkers due to its inherent flexibility and the lack of more available intact structures. Nevertheless, we currently assume that the cause of interactions between the condensing and the modifying regions could be contributed by multiple factors, e.g., the relative orientation between DH dimer and KS-AT domain (the conformation of the flexible AT-DH linker), the flexible DH-ψKR/KR linker and perhaps the interface constitutions of both DH-ψKR/KR linker and the LD-AT domains.

Figure_Response 4. Structural comparisons of the relative position between the condensing and the modifying regions. a, Superposition of mFAS and LovB structures on the KS domain of CalA3. The AT-DH linkers are highlighted without the rest of the structures showing the conformational differences.

b, Superposition of Alphafold2-predicted PpsD structure on the KS domain of CalA3 and on the DH domain of CalA3 (**c**). The AT-DH and DH-ψKR/KR linkers are highlighted without the rest of the structures showing the conformational differences.

****Figure_Response 4a-c are currently not incorporated in the manuscript (unless required to do so by the reviewer #2).****

12. Lines 132-134 - The proposed inactivation of the DH domain has not been confirmed using biochemical characterization. It is possible that Q962, E1129 and Q1133 can still maintain DH catalytic activity as the mutations are not vastly different from the typical residues: His, Asp and Gln (His). I suggest writing “likely nonenzymatic” or “likely inactive” instead of “nonenzymatic”.

A: Thank you for the suggestion! We have now used the term of “likely nonenzymatic” in the manuscript. Line 148: “Despite the topological similarity with other DH domains, the mutation of active site residues harbored by both hot-dog structures makes it the second likely nonenzymatic domain of CalA3...”

13. Since the cryoEM sample contained SNAC-N1 and 3-HAA, did the authors observe any map density for these compounds in the KS active site? Please also specify whether the map density of SNAC-N1 or 3-HAA was observed in the main text.

A: We did not observe clear and definitive map density for these compounds (Figure_Response 5). To be academically rigorous, we did not build any compounds but turned to seek assistance from the experiments of MD simulations and amino acid mutations to gain further insights into the catalytic mechanism.

We have now clearly stated in the main text that compounds were not observed.

Figure_Response 5. Close-up view of the substrate tunnels. The sharpened cryo-EM map of the substrate tunnel regions of CalA3 is contoured at 5σ . The arrows represent the direction of substrate entry of chain A (left) and chain B (right), respectively. The figure was generated in Pymol.

****Figure_Response 5 is currently not incorporated in the manuscript (unless required to do so by the reviewer #2).****

14. Please add labels for compounds corresponding to the peaks as SNAC-N1, hydrolysis product and amidation product in Fig. 3e and Extended data fig. 4.

A: We have now added labels for compounds as SNAC-N1, hydrolysis product and amidation product in Fig. 3e and Extended data Fig. 14 (Supplementary Fig. 15).

15. Have other PKSs having a domain organization like CalA3 (KS-AT-DH-KR and no ACP) been identified in other PKS pathways? Are the proposed inactivating features of the AT, DH, KR domains and the linker interactions that contribute to the rigid symmetric architecture of CalA3 conserved in any other PKSs?

A: To our best knowledge, no. The reviewer #2 raises a quite insightful question: to what extent does this rigid symmetric architecture of CalA3 conserved in the PKSs community. We think that the architecture of CalA3 is a rare occurrence yet quite special in the PKSs field for three reasons: 1. absence of ACP; 2. not a typical modular PKS which catalyzes Claisen condensation and beta-group modification; 3. rigid pseudo-twofold symmetry. As such, functional CalA3 may also be perceived as a gigantic type II enzyme for product release which distinguished CalA3 from canonical modPKS.

However, the CalA3 architecture suggests a possibility that the entire LD-AT-DH-ψKR/KR may have been evolved into a “king size lid”. Modular PKSs often contain inactive domains and some of them may have been evolved into lid-like domains. Similar cases were reported in other PKS systems. For example, in the curacin pathway, the loading module CurA contains MT_L-GNAT_L-ACP_L. The “MT_L” lost its ancestral function and was evolved into a lid-like subdomain and the “GNAT_L” is a catalytic decarboxylase subdomain. Thus, CurA “MT_L-GNAT_L” is in fact a “ψ decarboxylase/decarboxylase” domain. Therefore, the structure of CalA3 reported here may be another representative for the PKSs containing lid-like domains which serve as a structural stabilizing role.

Nevertheless, we believe that CalA3 structure adds to the growing, but still small, database of complete multi-domain PKS structures.

We have added the preceding paragraph in the Discussion section (Line 278-288).

Line 278: “The surrounding domains tightly contacting each other around the KS are unusual. The architecture suggests a possibility that the entire LD-AT-DH-ψKR/KR may have been evolved into a “king size lid”. These domains may lose canonical enzymatic function either by mutation or alteration during evolution and play only a structural stabilizing role. Similar case was reported in the curacin biosynthetic pathway¹. The loading module CurA contains methyltransferase-like (MT_L), GCN5-related N-acetyltransferase-like (GNAT_L) and loading-ACP domains (MT_L-GNAT_L-ACP_L). However, the “MT_L” lost its ancestral function and was evolved into a lid-like subdomain and the “GNAT_L” functions as a catalytic decarboxylase subdomain². Thus, CurA “MT_L-GNAT_L” is in fact a “ψ decarboxylase/decarboxylase” domain. Nevertheless, structural rearrangements may be a requirement for fully functional PKSs, as exemplified by reported megaenzymes and predicted structures.”

1. Gu, L. et al. GNAT-like strategy for polyketide chain initiation. *Science* **318**, 970–974 (2007).

2. Skiba, M. A. et al. Repurposing the GNAT fold in the initiation of polyketide biosynthesis. *Structure* **28**, 63-74 (2020).

Reviewer #3 (Remarks to the Author):

This manuscript reports the biochemical and structural characterization of an unusual polyketide synthase (PKS) module, CalA3, which is responsible for the amide bond formation in the biosynthesis of calcimycin. Initially, the authors revealed the involvement of CalA3 in the biosynthesis by gene deletion experiment, then obtained the cryo-EM structure, and finally performed *in vitro* enzymatic reactions, providing insight into how the unusual amide bond formation is catalyzed by CalA3. I found that the enzymatic reaction reported herein and the detailed structural analysis of CalA3 are of general interest; however, I would like to raise several concerns, which should be addressed before further consideration of the manuscript.

We are thankful for the reviewer's positive and professional comments on our work, especially the comments about enzymatic reactions which are very helpful to the quality promotion of this manuscript. The concerns have been carefully addressed point by point, the new mutation and truncation experiments have been performed for better supporting our conclusion, the figures have been added and revised as suggested and the manuscript has been extended to provide more information for readers and also revised for clarity.

1. I think that the Introduction section does not provide sufficient information for readers. Provide more information about calcimycin biosynthesis and explain why the biosynthesis is intriguing and should be studied. Also, it is not very clear why the authors suddenly mention the polyketide chain release by aminoacylation in lines 54-55. Some other minor issues are: (i) simvastatin is not a pure natural product (line 39); (ii) abbreviations, such as AT and DH, should be defined when they appear for the first time in the manuscript.

A: We have now provided more information about calcimycin biosynthesis and biochemical interest in the introduction section.

Line 48: "The polyether antibiotic calcimycin (A23187) in *Streptomyces chartreusis* NRRL 3882 is widely used as a biochemical tool in pharmacological and *in vitro* toxicological studies¹. As a divalent cation ionophore², calcimycin is also able to uncouple oxidative phosphorylation of mammalian cells³, inhibit ATPase activity⁴ and induce apoptosis via activation of intracellular signaling⁵. Structurally, calcimycin is composed of an α -ketopyrrole, a spiroketal ring (i.e., a polyketide) which is synthesized by the above-mentioned type I modular PKSs, and a substituted benzoxazole moiety which involves an unusual C-N bond formation and product release⁶ (Supplementary Fig. 1). Reported by X-ray crystallographic studies^{7,8}, the C-N bond contained benzoxazole system is crucial for calcium ion binding and thus indispensable for this ionophore-mediated biological processes. However, the mechanism by which this unique amidation reaction and chain release in calcimycin biosynthesis remains unknown."

1. H., B. J. & A., V. H. J. The use of the divalent calcium-ionophore A23187 as a biochemical tool in pharmacological and *in vitro* toxicological studies. *Cell Struct. Funct.* **21**, 97–99 (1996).

2. Pressman, B. C. Biological applications of ionophores. *Annu. Rev. Biochem.* **45**, 501–530 (1976).

3. Andreo, C. S. & Vallejos, R. H. Uncoupling of photophosphorylation in spinach chloroplasts by the ionophorous antibiotic A23187. *FEBS Lett.* **46**, 343–346 (1974).

4. Hara, H. & Kanazawa, T. Selective inhibition by ionophore A23187 of the enzyme isomerization in the catalytic cycle of sarcoplasmic reticulum Ca²⁺-ATPase. *J. Biol. Chem.* **261**, 16584–16590 (1986).

5. Kajitani, N. et al. Mechanism of A23187-induced apoptosis in HL-60 cells: dependency on mitochondrial permeability transition but not on NADPH oxidase. *Biosci. Biotechnol. Biochem.* **71**, 2701–2711 (2007).

6. Wu, Q. et al. Characterization of the biosynthesis gene cluster for the pyrrole polyether antibiotic calcimycin (A23187) in *Streptomyces chartreusis* NRRL 3882. *Antimicrob. Agents Chemother.* **55**, 974–982 (2011).

7. Chaney, M. O., Jones, N. D. & Debono, M. The structure of the calcium complex of A23187, a divalent cation ionophore antibiotic. *J. Antibiot.* **29**, 424–427 (1976).

8. Smith, G. D. & Duax, W. L. Crystal and molecular structure of the calcium ion complex of A23187. *J. Am. Chem. Soc.* **98**, 1578–1580 (1976).

(i) It has been revised as (Line 37): "(...and semi-synthetic simvastatin)".

(ii) We have now defined the abbreviations when they appear for the first time. Line 61: “However, as for the ketosynthase (KS)-acyltransferase (AT)-dehydratase (DH)-ketoreductase (KR)-acyl carrier protein (ACP) (i.e., KS-AT-DH-KR-ACP) domain organized modular PKSs...”.

Note: The abbreviation for each domain may also be found in the legend of Fig. 1.

2. The authors initially revealed that CalA3 is engaged in the biosynthesis by gene deletion experiment. This subsection is very briefly written, and it is not very clear why the authors focused only on CalA3 out of the enzymes encoded by the gene cluster. Also, the gene deletion experiment resulted in the elimination of calcimycin, but no accumulation of another metabolite was observed. Based on this result, I feel that it is difficult to predict that CalA3 is the enzyme for the polyketide chain release. Please provide a more reasonable explanation. In addition, please provide an image of the whole gene cluster in the supporting information.

A: Sorry for not providing suitable background for the reason why we focused only on CalA3. An image of the whole gene cluster has now been provided in Supplementary Fig. 1.

Figure_Response 1 (Supplementary Fig. 1). Calcimycin biosynthetic gene cluster in *Streptomyces chartreusis* NRRL 3882. a, Open reading frames (ORFs) of the calcimycin biosynthetic gene cluster. The prefix of “cal” was omitted on purpose from each gene. **b**, Structure of calcimycin.

Indeed, CalA3 was not on our radar initially. Starting from the characterization of calcimycin biosynthetic gene cluster by Wu et al. in 2011¹, we systematically investigate the cluster by performing multiple gene knockout experiments. Then, we *in vitro* reconstituted activities of several enzymes encoded by the gene cluster. For example: CalM is a N-methyltransferase that catalyze N-methyl modification on benzoxazole ring of calcimycin (Wu et al.²); CalR3 negatively controls calcimycin biosynthesis (Gou et al.³); CalC functions as an ATP-dependent CoA ligase that converts cezomycin to cezomycin-CoA to further calcimycin biosynthesis (Wu et al.⁴); and especially CalG, a type II thioesterase (TE) which was initially predicted to release the polyketide chain, turns out to be a dedicated TE that recycles overactivated acyls during calcimycin biosynthesis (Wu et al.⁵).

1. Wu, Q. et al. Characterization of the biosynthesis gene cluster for the pyrrole polyether antibiotic calcimycin (A23187) in *Streptomyces chartreusis* NRRL 3882. *Antimicrob. Agents Chemother.* **55**, 974–982 (2011).
2. Wu, Q. et al. Characterization of the N-methyltransferase CalM involved in calcimycin biosynthesis by *Streptomyces chartreusis* NRRL 3882. *Biochimie* **95**, 1487–1493 (2013).
3. Gou, L. et al. A novel TetR family transcriptional regulator, CalR3, negatively controls calcimycin biosynthesis in *Streptomyces chartreusis* NRRL 3882. *Front. Microbiol.* **8**, 2371 (2017).
4. Wu, H. et al. Cezomycin is activated by CalC to its ester form for further biosynthesis steps in the production of calcimycin in *Streptomyces chartreusis* NRRL 3882. *Appl. Environ. Microbiol.* **84**, e00586-18 (2018).
5. Wu, H. et al. Recycling of overactivated acyls by a type II thioesterase during calcimycin biosynthesis in *Streptomyces chartreusis* NRRL 3882. *Appl. Environ. Microbiol.* **84**, e00587-18 (2018).

How the C-N bond of benzoxazole ring is formed and the polyketide chain is released is a long-standing question. The gene knockout experiments showed that the phenotype of cezomycin and calcimycin elimination could only be caused by disruptions of two genes: *calU2* and *calA3*. However, our group previously showed that CalU2 was responsible for the spiroketal ring formation⁶ (Figure_Response 2, journal-unpublished). After ruling out the role of CalU2 and the above-mentioned CalG for the polyketide chain release, we were then promoted to propose that CalA3 is engaged in the C-N bond formation/chain release in calcimycin biosynthesis. Eventually, we provided the rigid symmetric CalA3 structure and the C-N bond formation/polyketide chain release function in this study.

Figure_Response 2. Functional analysis of CalU2⁶. HPLC traces showing the products of the *in vitro* reactions catalyzed by CalU2. The analysis indicates that CalU2 catalyzes spiroketal moiety formation.

6. Wang, X. Study on the function of C-skeleton biosynthesis related genes in calcimycin. (Shanghai Jiao Tong University, 2019).

Figure_Response 2 is journal-unpublished data which is only used here to explain to the reviewer #3.

The first subsection of the Results has been revised.

Line 76: “In calcimycin biosynthesis, a long-standing question is how the C-N bond of benzoxazole ring is formed and polyketide chain is released. Calcimycin is synthesized by modular CalA1, A2, A5 and A4¹, in which the domain organization matches well with the carbon backbone of the pyrrole 2-carboxylic acid-primed polyketide chain (Supplementary Fig. 2a). This chain reacts with 3-hydroxyanthranilic acid (3-HAA) in a hitherto unknown mechanism to form C-N bond in the benzoxazole ring system and released. Except CalG, a type II TE (thioesterase), which was initially predicted to release the polyketide chain but turns out to be a dedicated TE that recycles overactivated acyls during calcimycin biosynthesis², there are no other genes encoding a TE domain or a putative hydrolase in the cluster to release the polyketide chain...”

1. Wu, Q. et al. Characterization of the biosynthesis gene cluster for the pyrrole polyether antibiotic calcimycin (A23187) in *Streptomyces chartreusis* NRRL 3882. *Antimicrob. Agents Chemother.* **55**, 974–982 (2011).
2. Wu, H. et al. Recycling of overactivated acyls by a type II thioesterase during calcimycin biosynthesis in *Streptomyces chartreusis* NRRL 3882. *Appl. Environ. Microbiol.* **84**, e00587-18 (2018).

3. Related to the previous comment, it is not very clear to me why the authors first solved the structure of CalA3 before understanding the catalytic function of this enzyme.

A: We almost solved the structure and established the function of CalA3 at the fairly similar time scale. To be completely honest with the reviewer #3, our group has focused on analyzing catalytic functions of enzymes involved in calcimycin biosynthesis for a long time, but the structure of CalA3 is the first modular polyketide synthase, although noncanonical, that we solved ourselves. The rigid symmetric CalA3 structure really blew our mind, so we decided to depict the CalA3 structure first in the main text.

4. The authors mention that the noncatalytic domains contribute to the stabilization of the protein structure. However, this hypothesis was not experimentally investigated. The authors should perform *in vitro* enzymatic reactions using the standalone KS domain and some other truncated versions of CalA3 to further understand the roles of the noncatalytic domains.

A: We fully agree. In order to experimentally investigate the hypothesis that whether the noncatalytic domains contribute to the stabilization of the protein structure (i.e., whether the LD-AT and DH- ψ KR/KR domains are required for CalA3 function), we cloned, overexpressed and tried to purify the two truncated versions of CalA3 (i.e., KS domain alone and KS-LD-AT tri-domain). Unfortunately, none of them have been successfully purified due to protein insolubility or failure of overexpression (Figure_Response 3a-b). Furthermore, in order to test whether excised AT and ψ KR/KR domains would interact with each other (like the way the two domains interact in CalA3) and form a complex *in vitro*, we cloned AT and KR domains out of CalA3 individually and tried to incubate the two domains in solution. However, there was only ψ KR/KR domain that was successfully purified (Figure_Response 3c-d).

The CalA3 truncation experiments indicated that the deletion of either LD-AT-DH- ψ KR/KR domain or DH- ψ KR/KR domain destabilized the KS and disabled its function, hence we supposed that the noncatalytic domains (LD-AT-DH- ψ KR/KR) play a structure-stabilizing role and are most likely required for CalA3 function.

Figure_Response 3. SDS-PAGE analysis of truncated versions of CalA3. The targeted theoretical molecular weight of each truncation is labeled in **a**, KS domain; **b**, KS-LD-AT domain; **c**, AT domain; **d**, ψ KR/KR domain. Note that only ψ KR/KR domain is successfully purified. Abbreviations used for each gel: M, marker; T, total protein; S, supernatant; Ft, flow through; E_Ni, elution of His-tagged protein; E_Strep, elution of StrepII-tagged protein, SEC, size exclusion chromatography.

Figure_Response 3a-d are currently not incorporated in the manuscript (unless required to do so by the reviewer #3).

5. CalA3 can hydrolyze compound 1 to form 2, and the authors state that the hydrolysis might be “due to the open-ended tunnel of KS” (lines 171-172). The catalytic mechanism of this hydrolysis is not very clear, although it seems the catalytic residues of the KS are not used for the reaction. Please provide a more detailed explanation for this chemical reaction.

A: Yes, we agree with the reviewer’s concern about the catalytic mechanism of hydrolysis. In order to pinpoint the residues specifically for hydrolysis, we performed double-point mutations (C197A+H332A, C197A+H372A, H332A+H372A) among the active site residues. The overexpression and purification of each double-point CalA3 mutant was hardly successful and the hydrolysis activity was not abolished (Figure_Response 4). Moreover, we kept on performing single-point mutations of residues lining across the catalytic tunnel of CalA3 KS domain and purified most of them successfully (Figure_Response 5). However,

HPLC analyses showed that none of the mutants eliminated the hydrolysis product (HPLC traces not shown). The results precluded us from further proposing a catalytic mechanism of this hydrolysis.

To be academically rigorous, we are sorry to say that we are currently unable to further explain this chemical reaction of hydrolysis in detail.

Figure_Response 4. Analysis of the hydrolysis activity of the CalA3 double-point mutants. Left, SDS-PAGE analysis of the double-point mutants of CalA3. Right, HPLC traces showing the products of the *in vitro* reactions catalyzed by CalA3 wild type (WT) and three mutants.

Figure_Response 5. SDS-PAGE analysis of the CalA3 mutants. The residues which line across the substrate tunnel were selected to mutate. Abbreviations used for each gel: M, marker; T, total protein; S, supernatant; P, pellet; Ft, flow through; E_Ni, elution of His-tagged protein; Ft_H, flow through of heparin column; E_H, elution of heparin column; SEC, size exclusion chromatography, Con, protein concentration.

****Figure_Response 4-5 are currently not incorporated in the manuscript (unless required to do so by the reviewer #3).****

6. This manuscript lacks a discussion on the comparison of the CalA3-catalyzed reaction and other noncanonical reactions catalyzed by KS-like enzymes. Especially, the authors' group previously discovered KS-catalyzed amide bond formation in the biosynthesis of A33853 (Chem. Biol., 2015, 22, 1313-1324), but this work is not cited in this manuscript. Please emphasize the novelty of the CalA3-catalyzed reaction in comparison with similar preceding cases.

A: Sorry for missing the excellent paper (*Chem. Biol.* **22**, 1313-1324 (2015)). We have now briefly compared CalA3-catalyzed reaction with other noncanonical reactions by KS-like enzymes and emphasized the novelty of CalA3 in the Discussion section.

Line 257: “The KS domain is the most highly conserved domain in either the architecture or active site residues. Indeed, there are limited KS domains which have been reported to catalyze noncanonical reactions. NonJ and NonK are highly homologous to known KSs and they catalyze the formation of C-O bond in nonactin biosynthesis rather than the canonical C-C bond¹. CerJ is also a KS homolog which catalyzes the C-O bond formation in cervimycin biosynthesis². KS domain in TAS1 which is a NRPS-PKS hybrid enzyme is involved in the cyclization step for TeA release³. KS domain in branching module of RhiE catalyzes vinylogous chain branching in rhizoxin biosynthesis⁴⁻⁵. In particular, the function of an unusual KS-like enzyme, BomK, was firmly established by Lv et al⁶. BomK is responsible for amide bond formation in antibiotic A33853 biosynthesis which provides the first clue for the KS-catalyzed C-N bond formation during the biosynthesis of benzoxazole-contained natural products. Nevertheless, by combining structural characterization using cryo-EM with biochemical characterization and MD simulations, the CalA3 KS-catalyzed C-N bond formation and chain release improved our understanding of functional divergence of apparent KS domains. Considering that the LD-AT-DH-ψKR/KR domains of CalA3 are likely nonenzymatic, CalA3 KS may also be perceived as a terminal amidase or hydroxylase and functional CalA3 may be viewed as a gigantic type II enzyme for product release which distinguished CalA3 from the above-mentioned PKSs. Moreover, the structure of CalA3 KS adopts the typical C-H-H active site residues, suggesting no major catalytic core reconfiguration. Given the similarity between the mechanism of KS and TE (both employing a Ser/Cys nucleophile and a basic residue), the repurposed KS domain of CalA3 reflects its own catalytic potential.

1. Kwon, H.-J. et al. C-O bond formation by polyketide synthases. *Science* **297**, 1327–1330 (2002).

2. Bretschneider, T. et al. A ketosynthase homolog uses malonyl units to form esters in cervimycin biosynthesis. *Nat. Chem. Biol.* **8**, 154–161 (2012).

3. Yun, C.-S., Motoyama, T. & Osada, H. Biosynthesis of the mycotoxin tenuazonic acid by a fungal NRPS–PKS hybrid enzyme. *Nat. Commun.* **6**, 8758 (2015).

4. Bretschneider, T. et al. Vinylogous chain branching catalysed by a dedicated polyketide synthase module. *Nature* **502**, 124–128 (2013).

5. Sundaram, S., Heine, D. & Hertweck, C. Polyketide synthase chimeras reveal key role of ketosynthase domain in chain branching. *Nat. Chem. Biol.* **11**, 949–951 (2015).

6. Lv, M., Zhao, J., Deng, Z. & Yu, Y. Characterization of the biosynthetic gene cluster for benzoxazole antibiotics A33853 reveals unusual assembly logic. *Chem. Biol.* **22**, 1313–1324 (2015).

Other minor points:

1. Line 164 – Does SNAC-N1 correspond to compound 1? Please clarify.

A: Yes, SNAC-N1 corresponds to compound 1 and sorry for the unclarity. We have now revised this sentence as follows (Line 181): “...we synthesized SNAC-N1 (Fig. 3e, compound 1), the substrate analog, from 11-oxo-11-(pyrrol-2-yl) undecanoic acid and SNAC (Supplementary Fig. 23).”

For clarify, we have also added labels for compounds 1, 2 and 3 as SNAC-N1, hydrolysis product and amidation product in Fig. 3e.

2. Fig. 3f – Some arrows and charges are missing in the figure.

A: Did you mean Fig. 3h, i.e., the proposed reaction mechanism figure? We have now revised this figure and please let us know if there are still issues in this panel.

3. References are not provided in a unified manner.

A: We have now corrected the formatting issues in the References section and provided it in a unified *Nature Communications* manner.

4. Extended Data Fig. 1 – “H20” should be a typo of “H₂O”.

A: Sorry for the careless. We have now revised it as “H₂O” in Supplementary Fig. 2.

REVIEWER COMMENTS

Reviewer #1 (Remarks to the Author):

After revision, the quality of this manuscript was largely improved. The authors experimentally proved that CalA3 KS catalyzes chain termination via amidation/hydrolysis and other CalA3 domains likely play a critical structural role. The authors also clearly stated that CalA3 is not a typical PKS module and revised the manuscript accordingly.

My major concern is the novelty and general impact of this study: CalA3 is a non-canonical PKS module, and the authors failed to identify any other modular PKS with a (proposed) similar termination strategy. The structural discovery of a pseudo two-fold symmetry/rigidity is quite interesting; however, if CalA3 remains the only known "modular" PKS with a unique "KS-AT-DH-KR" architecture for chain release, I am doubtful that this structural discovery should be included in the general discussion of importance of PKS structural asymmetry/flexibility to its function.

From my perspective, the focus of this study should be structural basis for KS-catalyzed chain termination in calcimycin biosynthesis. The authors performed mutagenesis and MD simulations to understand CalA3 catalysis, but I think it is important to obtain a substrate or product complex structure for the publication of this study in Nature Communications. The authors did not observe definitive electron density maps for either substrate and moved forward with docking models, which contributes to less convincing biocatalytic conclusions. Visualizing substrate or product will verify subsequent analysis of the binding mode from experimentally determined maps. I would recommend trying to prepare cryo-EM samples with the amidation product 3 or the hydrolysis product 2 (preferably 3). The authors should also include Figure_Response 1 in the manuscript for publication.

Reviewer #2 (Remarks to the Author):

Overall I think the authors have addressed all of our comments. I suggest adding another cryo-EM map figure for CalA3 DD domain in the supplementary materials. Since the models shown in figures 1 and 2 include the DDs, and the second DD conformation flanking the AT domain has not been observed before, including the map density for these regions in the supplementary document will be helpful for the readers.

Reviewer #3 (Remarks to the Author):

I appreciate the authors' effort to revise the manuscript, which I found has been improved. I only have a few comments/requests, as shown below.

- In the "CalA3 is involved in polyketide chain release in calcimycin synthesis" section, please add the background information as the authors mentioned in the response letter.
- Lines 88-89: Please move this sentence ("To gain further insights...") to the beginning of the next section.
- Fig. 3h: (1) In the first step of the reaction, one of the arrows (which is going to the sulfur atom) should go to the carbonyl oxygen. (2) The negative charge on the oxygen atom is missing in structures at the upper and lower right. (3) In the third step of the reaction, there needs to be an additional arrow from the oxygen atom.

Point-by-point response to the reviewers (second round)

*Reviewer Comments: Black, Helvetica, 10

*Author Responses: Blue, Helvetica, 10

Note: "A:" stands for "Answer:".

Reviewer #1 (Remarks to the Author):

After revision, the quality of this manuscript was largely improved. The authors experimentally proved that CalA3 KS catalyzes chain termination via amidation/hydrolysis and other CalA3 domains likely play a critical structural role. The authors also clearly stated that CalA3 is not a typical PKS module and revised the manuscript accordingly.

My major concern is the novelty and general impact of this study: CalA3 is a non-canonical PKS module, and the authors failed to identify any other modular PKS with a (proposed) similar termination strategy. The structural discovery of a pseudo two-fold symmetry/rigidity is quite interesting; however, if CalA3 remains the only known "modular" PKS with a unique "KS-AT-DH-KR" architecture for chain release, I am doubtful that this structural discovery should be included in the general discussion of importance of PKS structural asymmetry/flexibility to its function.

Again, we thank the reviewer #1 for your recognition of our revised manuscript and the critical comment about the exceptionality of CalA3 structure.

From my perspective, the focus of this study should be structural basis for KS-catalyzed chain termination in calcimycin biosynthesis. The authors performed mutagenesis and MD simulations to understand CalA3 catalysis, but I think it is important to obtain a substrate or product complex structure for the publication of this study in Nature Communications. The authors did not observe definitive electron density maps for either substrate and moved forward with docking models, which contributes to less convincing biocatalytic conclusions. Visualizing substrate or product will verify subsequent analysis of the binding mode from experimentally determined maps. I would recommend trying to prepare cryo-EM samples with the amidation product 3 or the hydrolysis product 2 (preferably 3). The authors should also include Figure_Response 1 in the manuscript for publication.

A: To be honest with the reviewer #1, we did not believe that we would observe a CalA3 structure complexed with any chemical compound. Before this manuscript was drafted in the first place, we had tried many approaches to obtain such a complex structure, but none of them were successful.

However, there is an idiom in Chinese called "先见之明", which means that your recommendation is impressive and foresighted. Only followed your suggestion were we finally able to obtain the amidation product and the hydrolysis product complex structures of CalA3 (Fig. 4b, d, e, f). We would like to express our gratitude to the reviewer #1 for the recommendation of preparing cryo-EM samples with products. Figure_Response 1 of the first revision round has been now included in the manuscript as Supplementary Fig. 15.

Line 240-267: "To confirm the binding mode predicted by the MD simulations, we tried to prepare cryo-EM samples of CalA3 with the amidation product 3 and the hydrolysis product 2, respectively, and the two complex structures were finally obtained. The cryo-EM map of CalA3 with the amidation product was resolved at overall resolution of 3.84 Å and the amidation products were captured in the KS domains of both chain A and B (Fig. 4b, Supplementary Figs. 21 and 23). The benzoxazole moiety may undergo hydrophobic interactions with A336 and L439 and form additional hydrogen bond with R1718 which is at a distance of 3.5 Å (Fig. 4d). The polyketide chain and the pyrrole ring could also be positioned by hydrophobic interactions contributed by F236 and Q147 respectively. The cryo-EM map of CalA3 with the hydrolysis product was resolved at overall resolution of 3.97 Å and the hydrolysis product was captured

only in the KS domain of chain A (Fig. 4b, Supplementary Figs. 22 and 23). The carboxyl oxygens may form hydrogen bonds with C197, T438 and L439 (Fig. 4e). Additionally, the carboxyl group of the hydrolysis product could also form a salt bridge with H372 at a distance of 5.4 Å which also demonstrates the positioning function of H372 predicted by MD simulations. Furthermore, we superposed the experimentally observed KS domain structures that contain the amidation and hydrolysis products respectively on the MD simulated KS domain structure that contains SNAC-N1 and 3-HAA to compare the relative locations of the substrates and the products (Fig. 4f). The C-N bond-formed benzoxazole moiety is shifted approximately 3.0 Å and rotated $\sim 51.5^\circ$ from the non-covalent 3-HAA substrate. The polyketide chain moieties are subtly moved within the range of 0.6 to 2.2 Å between the SNAC-N1, amidation and hydrolysis products. The conformational variability of the pyrrole rings of the three compounds, however, is relatively significant. To consider the pyrrole and the carbonyl group of each chain as one unit, it is shifted ~ 6.5 Å and rotated $\sim 53.6^\circ$ from SNAC-N1 to the hydrolysis product, which the latter is able to further translate ~ 6.4 Å and rotate $\sim 74.4^\circ$ compared to the amidation product. Similarly, a motion of this unit combining movement of ~ 5.8 Å with rotation of 34.9° is also observed compared between the amidation product and SNAC-N1. Together, the visualization of the products in the KS domain verifies the basic binding location in the substrate tunnel analyzed by MD simulations and further reveals the structural flexibility of the polyketide chain."

** In this response letter, we first show the result (i.e., Fig. 4) in the next page. Then, the corresponding figures, experimental processes, descriptions and methods are supplemented after the result. **

** Any figures, descriptions and methods involving cryo-EM analysis of CalA3 with SNAC-pyrrol-propanethioate+3-HAA (discussed later) in this response letter are currently not included in the manuscript (unless required to do so by the reviewer #1). **

Figure_Response_{second round} 1 (Fig. 4). Catalytic mechanism of CalA3 KS domain. **a** Coulombic electrostatic potential (ESP) surface representation of KS domain, docked with 3-HAA and SNAC-N1 in the substrate tunnel. **b** Surface representation of KS domains captured with the amidation and hydrolysis products in the substrate tunnels respectively. **c** MD simulation of the covalently acylated polyketide chain (SNAC-N1) and the noncovalently docked 3-HAA in the active site region of KS domain. The substrates are coordinated by catalytic and positioning residues, which are marked with asterisk and hashtag, respectively. The distances between them are labeled (Å). Detailed views of the amidation product (**d**) and the hydrolysis product (**e**) binding in the active site region of KS domain respectively. The cryo-EM map for

each product is shown as mesh and carved at 2.5 Å distance. The map is contoured at 9.0 σ for the amidation and at 7.6 σ for the hydrolysis product, respectively. **f** Superposition of the products captured KS domains with MD simulated KS showing the conformational variability of each ligand. **g** Proposed amidation reaction mechanism.

After accepting the suggestion of the reviewer #1, we tried to prepare three sets of cryo-EM samples: ① CalA3+amidation product+3-HAA; ② CalA3+hydrolysis product+3-HAA; ③ CalA3+SNAC-pyrrol-propanethioate+3-HAA.

The reason why we temporarily introduced ③ into the system was because we supposed that a non-efficient substrate may be more stabilized than SNAC-N1 which could be better captured by cryo-EM analysis. We synthesized the SNAC-pyrrol-propanethioate from 3-(1*H*-pyrrol-2-yl) propanoic acid. The carbon chain length of this substrate is much shorter compared to SNAC-N1. We incubated CalA3 with SNAC-pyrrol-propanethioate and 3-HAA. Although less efficient, the amidation product formed between SNAC-pyrrol-propanethioate and 3-HAA was confirmed by HPLC and LC-MS (Figure_Response_{second round} 2). The hydrolysis product was non-detectable in this reaction. The cryo-EM map was resolved at overall resolution of 3.79 Å (Figure_Response_{second round} 5), however, we did NOT observe definitive cryo-EM map for either SNAC-pyrrol-propanethioate and 3-HAA (Figure_Response_{second round} 6).

The cryo-EM sample preparations of ① and ② were strikingly successfully. The cryo-EM maps of ① and ② were resolved at overall resolution of 3.84 Å and 3.97 Å (Figure_Response_{second round} 3-4), respectively. The amidation and hydrolysis products were captured respectively (Figure_Response_{second round} 6). We analyzed the binding locations of the amidation product and the hydrolysis product and compared them with the MD simulated SNAC-N1 and 3-HAA. The subtle motion between them confirmed that the MD simulations were reasonable.

Figure_Response_{second round} 2. C-N bond formation by CalA3 between SNAC-pyrrol-propanethioate and 3-HAA. **a** Schematic representation of the amidation reaction. **b** HPLC traces at $\lambda=254$ nm showing the amidation product of the *in vitro* reaction. **c-d** Representative extracted ion chromatogram traces in positive ion mode showing the amidation product of the *in vitro* reaction. The selective m/z $[M+H]^+$ value used to detect the product is labeled over the trace, and the amidation product has the calculated m/z $[M+H]^+$ value of 275.1026.

Figure_Response_{second round} 3 (Supplementary Fig. 21). Cryo-EM structure determination of CalA3 with the amidation product. **a** One representative cryo-EM micrograph from the 2516 movie stacks of CalA3 with selected particles in white circles. Scale bar, 20 nm. **b** Representative 2D class averages. **c** A data processing workflow for the resolution-labeled cryo-EM maps. **d** Local resolution of the map estimated in RELION. **e** Angular distribution of all particles used for the final reconstruction of the map. **f** Gold-standard FSC curves of the resolution-labeled cryo-EM map (FSC=0.143 criterion). **g** FSC curves of the final refined model versus the map that it was refined against (black); of the model refined in the first of the two independent maps used for the gold-standard FSC versus that same map (blue); and of the model refined

in the first of the two independent maps versus the second independent map (red). The small difference between the work and free FSC curves indicates that the model did not suffer from overfitting.

Figure_Response_{second round} 4 (Supplementary Fig. 22). Cryo-EM structure determination of CalA3 with the hydrolysis product. a One representative cryo-EM micrograph from the 2119 movie stacks of CalA3 with selected particles in white circles. Scale bar, 20 nm. **b** Representative 2D class averages. **c** A data processing workflow for the resolution-labeled cryo-EM maps. **d** Local resolution of the map estimated in RELION. **e** Angular distribution of all particles used for the final reconstruction of the map. **f** Gold-standard FSC curves of the resolution-labeled cryo-EM map (FSC=0.143 criterion). **g** FSC curves of the final refined model versus the map that it was refined against (black); of the model refined in the first of the two independent maps used for the gold-standard FSC versus that same map (blue); and of the model refined in the first of the two independent maps versus the second independent map (red). The small difference between the work and free FSC curves indicates that the model did not suffer from overfitting.

Figure_Response_{second round} 5. Cryo-EM structure determination of CalA3 with SNAC-pyrrol-propanethioate and 3-HAA. **a** One representative cryo-EM micrograph from the 2280 movie stacks of CalA3 with selected particles in white circles. Scale bar, 20 nm. **b** Representative 2D class averages. **c** A data processing workflow for the resolution-labeled cryo-EM maps. **d** Local resolution of the map estimated in RELION. **e** Angular distribution of all particles used for the final reconstruction of the map. **f** Gold-standard FSC curves of the resolution-labeled cryo-EM map (FSC=0.143 criterion). **g** FSC curves of the final refined model versus the map that it was refined against (black); of the model refined in the first of the two independent maps used for the gold-standard FSC versus that same map (blue); and of the model refined in the first of the two independent maps versus the second independent map (red). The small difference between the work and free FSC curves indicates that the model did not suffer from overfitting.

Figure_Response_{second round} 6 (Supplementary Fig. 23). Close-up view of the substrate tunnels of CalA3 KS domains. The sharpened cryo-EM maps of the substrate tunnel regions of CalA3 are shown as mesh. The map incubated with the amidation products of chain A and B is contoured at 10.3σ and 7.9σ respectively. The map incubated with the hydrolysis product of chain A and B is contoured at 7.7σ and 10.6σ respectively. The map incubated with SNAC-pyrrol-propanethioate and 3-HAA of chain A and B is contoured at 8.8σ . The solid arrows point at the map for each product. The dotted arrows represent the direction of substrate entry of chain A (left) and chain B (right), respectively, without observing the ligands. The figure was generated in Pymol. The cryo-EM maps of ligands, shown on the right as surfaces, are contoured at 5.6σ and 6.5σ for the amidation products of chain A and B respectively, and at 6.3σ for the hydrolysis product of chain A and generated in ChimeraX. The carving distance is 2.5 \AA .

Methods:

***In vitro* reconstitution experiment of CalA3 on SNAC-pyrrol-propanethioate**

5 μM CalA3 WT were incubated with 1 mM 3HA and 1 mM *S*-(2-acetamidoethyl)3-(1*H*-pyrrol-2-yl)propanethioate in buffer (25 mM Tricine pH 7.7) in a 100 μL volume at 30 $^{\circ}\text{C}$ for 14 h. The reactions were quenched with 100 μL methanol. The precipitated protein was removed after centrifugation at $15,000 \times g$ for 20 min and the supernatant was prepared for HPLC and LC-MS analysis.

HPLC analysis was conducted with an Agilent C18 reverse-phase column (Pursuit XRs C18, 3 μm , 250 \times 4.6mm; Agilent) on an Agilent HPLC system (Agilent 1260 Infinity; Agilent) at a flow rate of 0.4 mL/min, a sample injection volume of 10 μL and mobile phase A (water/0.1% formic acid) and B (methanol/0.1% formic acid). Mobile A was gradually replaced by 75% to 85% volumes of mobile B over a period of 10 to

18 min, 85% to 95% volumes for 18 to 32 min, 95% volumes for 32 to 35 min. The elution was monitored by UV spectroscopy at λ 254 nm and the amount of product present in the reaction mixtures was calculated from the peak area by Agilent OpenLAB CDS ChemStation Edition software.

LC-MS analysis was conducted with an Agilent C18 reverse-phase column (Pursuit XRs C18, 3 μ m, 250 \times 4.6mm; Agilent) on an Agilent 1290 Infinity Liquid chromatography and 6545 Quadrupole Time-of-Flight Mass Spectrometer using positive electrospray ionization. Samples were eluted at a flow rate of 0.4 mL/min with a sample injection volume of 10 μ L and the same mobile phases as HPLC analysis. Mobile A was gradually replaced by 75% to 85% volumes of 0 to 8 min, 85% to 95% volumes for 8 min to 22 min, 95% to 100 % volumes for 22 min to 29 min, 100% volumes for 29 min to 35 min, and 100% to 75% volumes for 35 min to 40 min. Product masses was determined offline using ESI-MS in positive-ion mode and the mass spectrometer was set to acquire spectra in the mass range 50-1000 m/z.

Cryo-EM specimen preparation, data acquisition and image processing

Three sets of cryo-EM specimens were prepared: (1) CalA3+amidation product+3-HAA; (2) CalA3+hydrolysis product+3-HAA; (3) CalA3+SNAC-pyrrol-propanethioate+3-HAA. For all three sets, four-microliter aliquots of specimens at \sim 4.5 mg/mL were applied to glow-discharged holey carbon grids (C-Flat Au, R1.2/1.3, 300 mesh) for 6 s of incubation, blotted for 2.5 s and plunge-frozen into liquid ethane precooled by liquid nitrogen using a Vitrobot Mark IV (FEI) operated at approximately 100% humidity and 4.5 $^{\circ}$ C. Cryo-EM images were collected with a Titan Krios electron microscope (FEI) operated at 300 kV and equipped with a K3 Summit direct electron detector (Gatan). Forty frames were recorded for each movie stack at a nominal magnification of 81000-fold in super-resolution mode with a pixel size of 0.89 \AA within the defocus range of 1.0 to 2.0 μ m. A total of 2516, 2280, and 2119 movie stacks of CalA3 dataset (1), (2) and (3), respectively, were automatically collected using EPU software with an exposure time of 2.67 s (0.1 s per frame) and a total dose of 48.06 e-/ \AA^2 .

The similar workflow was used to process all three datasets separately. All movies of the datasets were aligned and dose-weighted using MotionCor2. The contrast transfer function (CTF) and defocus parameters were determined by CTFFIND 4¹. Micrograph checking, particle autopicking, 2D and 3D classification, autorefinement, postprocessing and resolution estimation of each cryo-EM map were performed using RELION 4.0². Approximately 2,000 particles of each dataset were manually picked and subjected to reference-free 2D classification. The best representative 2D classes were selected as templates for autopicking.

For the reconstructions of CalA3, the datasets were cleaned by removing ice contaminants and junk particles after two rounds of 2D classification, and good classes were kept to generate the 30- \AA 3D initial model, which was low-pass filtered to 70 \AA as the reference for subsequent 3D classification. The best classes of 223,652, 141,918, 141,872 particles for (1), (2) and (3) were selected for autorefinement, which resulted in the reconstructions of 4.16, 4.38, and 4.02- \AA CalA3 consensus cryo-EM maps, respectively. Subsequent CTF refinement, Bayesian polishing, autorefinement and postprocessing were performed, yielding the 3.84, 3.97, and 3.79 \AA cryo-EM map imposed with C1 symmetry (Supplementary Figs. 21-22).

The resolutions of all cryo-EM maps were estimated based on the corrected gold standard Fourier shell correlation (FSC) at the 0.143 criterion (Supplementary Figs. 21f and 22f).

Model refinements and ligands building of the amidation and hydrolysis products were conducted using ISOLDE based on the CalA3 atomic model.

1. Rohou, A. & Grigorieff, N. CTFFIND4: Fast and accurate defocus estimation from electron micrographs. *J. Struct. Biol.* **192**, 216–221 (2015).

2. Kimanius, D., Dong, L., Sharov, G., Nakane, T. & Scheres, S. H. W. New tools for automated cryo-EM single-particle analysis in RELION-4.0. *Biochem. J.* **478**, 4169–4185 (2021).

Reviewer #2 (Remarks to the Author):

Overall I think the authors have addressed all of our comments. I suggest adding another cryo-EM map figure for CalA3 DD domain in the supplementary materials. Since the models shown in figures 1 and 2 include the DDs, and the second DD conformation flanking the AT domain has not been observed before,

including the map density for these regions in the supplementary document will be helpful for the readers.

We would like once again to express our appreciation to the reviewer #2 for evaluating our work. The cryo-EM map for CalA3 DD domain has now been added as Supplementary Fig. 11.

Figure_Response_{second round} 1 (Supplementary Fig. 11) The cryo-EM maps of CalA3 N-terminal docking domains (DD). Left, flanked DD domain, unsharpened map contoured at 0.039 (1.6 σ) and DeepEMhancer sharpened map contoured at 0.003 (0.07 σ); right, perpendicular DD domain, unsharpened map contoured at 0.0067 (3.4 σ) and DeepEMhancer sharpened map contoured at 0.09 (2.1 σ).

Reviewer #3 (Remarks to the Author):

I appreciate the authors' effort to revise the manuscript, which I found has been improved. I only have a few comments/requests, as shown below.

We are truly happy for the reviewer's positive recognition of our work and thankful for your patience to advise us about the mechanism figure.

- In the "CalA3 is involved in polyketide chain release in calcimycin synthesis" section, please add the background information as the authors mentioned in the response letter.

A: We have now added the previously mentioned background information in the section from line 84 to line 99.

- Lines 88-89: Please move this sentence ("To gain further insights...") to the beginning of the next section.

A: This sentence has now been moved to the beginning of the "Overall architecture of CalA3" section (Line 105).

- Fig. 3h: (1) In the first step of the reaction, one of the arrows (which is going to the sulfur atom) should go to the carbonyl oxygen. (2) The negative charge on the oxygen atom is missing in structures at the upper and lower right. (3) In the third step of the reaction, there needs to be an additional arrow from the oxygen atom.

A: Thanks! The figure has been corrected and revised.

REVIEWERS' COMMENTS

Reviewer #1 (Remarks to the Author):

The authors have addressed all of the reviewers' comments, and analysis of the product complex structures nicely contributed to the overall understanding of this unique KS-catalyzed termination reaction. I recommend publication of this study in Nature Communications.

Point-by-point response to the reviewers (third round)

Reviewer #1 (Remarks to the Author):

The authors have addressed all of the reviewers' comments, and analysis of the product complex structures nicely contributed to the overall understanding of this unique KS-catalyzed termination reaction. I recommend publication of this study in Nature Communications.

Thanks!